


# The potential of big data and machine learning for weather index insurance

Luigi Cesarini[1], Rui Figueiredo[2], Beatrice Monteleone[1], and Mario Lloyd Virgilio Martina[1]

[1]Scuola Universitaria Superiore IUSS Pavia, Pavia, 27100, Italy
[2]CONSTRUCT-LESE, Faculty of Engineering, University of Porto, 4200-465 Porto, Portugal

**Correspondence:** Luigi Cesarini (luigi.cesarini@iusspavia.it)

**Abstract.** Weather index insurance is an innovative tool in risk transfer for disasters induced by natural hazards. This paper proposes a methodology that uses machine learning algorithms for the identification of extreme flood and drought events aimed at reducing the basis risk connected to this kind of insurance mechanism. The model types selected for this study were the neural network and the support vector machine, vastly adopted for classification problems, which were built exploring thousands of

possible configurations based on the combination of different model parameters. The models were developed and tested in the Dominican Republic context, based on data from multiple sources covering a time period between 2000 and 2019. Using rainfall and soil moisture data, the machine learning algorithms provided a strong improvement when compared to logistic regression models, used as a baseline for both hazards. Furthermore, increasing the number of information provided during the training of the models proved to be beneficial to the performances, increasing their classification accuracy and confirming the

ability of these algorithms to exploit big data and their potential for application within index insurance products.

## 1   Introduction

Changes in frequency and severity of extreme weather and climate events have been observed since 1950, including an increase in the number of heavy precipitation events in some land areas and a significant decrease of rainfall in other regions (Field et al.,

2014). Impacts from recent weather-related extremes, such as floods and droughts, have revealed a significant vulnerability of many human systems to climate-related hazards (Visser et al., 2014). In recent decades, extreme weather events have caused widespread economic and social damages all over the world (Kron et al., 2019). According to Hoeppe (2016), over the period from 1980 to 2014, extreme weather events have caused losses of around US$ 3.3 billion, with floods accounting for 32% of the losses and drought for 17%. Extreme weather events have devastating effects on people's lives as well. The International

Disasters Database EMDAT (CRED, 2019) reports that, over the period from 1980 to 2019, extreme weather caused the death of 1.15 million people, with droughts being the disaster responsible for the highest number of deaths (around 50% of fatalities due to climate extremes), followed by storms (34%) and floods (16%).





The implementation of effective disaster risk management strategies is key to limiting economic and social losses associated with extreme weather events and to reduce disaster risk. In recent years, there has been increasing worldwide interest in the integration of risk transfer instruments within such strategies (Kunreuther, 2001; Surminski et al., 2016). Among those instruments, index-based insurance, or parametric insurance, has gained remarkable popularity. Unlike traditional insurance, which indemnifies policyholders based on experienced losses, parametric insurance pays indemnities based on realizations of an index (or a combination of parameters) that is correlated with losses (Barnett and Mahul, 2007). It can be used to transfer risk associated with different types of extreme events, such as earthquakes (Franco, 2010), floods (Surminski and Oramas-Dorta, 2014), and droughts (Makaudze and Miranda, 2010). Parametric insurance offers various advantages over traditional indemnity-based insurance, such as lower operating expenses, reduced moral hazard and adverse selection, and prompt access to funds by insureds following the occurrence of disasters (Ibarra and Skees, 2007; Figueiredo et al., 2018). This promptness is critical in developing countries, which tend to be exposed to short-term liquidity gaps that may overwhelm their capacity to cope with large disasters (Van Nostrand and Nevius, 2011). A critical disadvantage of parametric insurance, however, is its susceptibility to basis risk, which may be defined as the risk that triggered payouts do not coincide with the occurrence of loss events.

The minimization of basis risk in parametric insurance requires a reliable, rapid and objective identification of extreme climate events. Nowadays, different sources of weather data that may be used to support this endeavour are available. Among them, the use of satellite images and reanalysis products in parametric insurance mechanisms is growing (Black et al., 2016; Chantarat et al., 2013). Satellite images and reanalysis are frequently free of charge, and therefore parametric models based on them are cheaper and can be affordable even for developing countries (Castillo et al., 2016). In addition, satellite images and reanalysis consist of continuous spatial fields and often have global coverage. These last features make them attractive, since they overcome one of the most common issues related to gauges and weather stations, which is their limited or irregular spatial coverage. It should also be noted that hypothetically, if an entity that is responsible for such stations (e.g. a governmental agency) is related in some form with a potential beneficiary from index insurance coverage, a conflict of interest may arise. This issue is avoided with satellite-based or reanalysis products, which are produced by third parties, for example internationally renowned research institutes such as the Climate Hazards Center of the University of California and the European Centre for Medium-Range Weather Forecasts. Satellite images are often available with high spatial resolution, but records are still short, with a maximum duration of around 30 years. Reanalysis, on the other hand, provides longer time series but tends to have a coarser spatial resolution. The combined use of various sources of information to detect the occurrence of extreme events is valuable, since it can significantly improve the ability to correctly detect extreme events (Chiang et al., 2007). However, the main challenge associated with the use of these sources of information is the management of such a vast amount of data (Chen et al., 2019).

Over the last two decades there has been an increasing attention to the application of machine learning methods to process and extract information from big data with limited human intervention (Ornella et al., 2019). Correspondingly, machine learning approaches have also been applied to forecast extreme events. Mosavi et al. (2018) offer an accurate description of the state of the art of machine learning models used to forecast floods, while Hao et al. (2018) and Fung et al. (2019), in their reviews





on drought forecasting, give an overview of machine learning tools applied to predict drought indices. Machine learning has also been employed to forecast wind gusts (Sallis et al., 2011), severe hail (Gagne et al., 2017) and excessive rainfall (Nayak and Ghosh, 2013). In contrast, only a minor part of the body of literature focuses their attention on the identification or classification of events (Nayak and Ghosh (2013), Khalaf et al. (2018) and Alipour et al. (2020) for floods, Richman et al. (2016) for droughts, Kim et al. (2019) for tropical cyclones). However, classification of events to distinguish between extreme and non-extreme events is essential to support the development of effective parametric risk transfer instruments. In addition, the major part of the analysed studies deals with a single type of event.

This paper aims to assess the potential of machine learning for weather index insurance. To achieve this, we propose and apply a machine learning methodology that is capable of objectively identifying and classifying extreme weather events in near-real time, regardless of the type of event. This methodology is then used to address the following research questions:

1. Can machine learning algorithms provide improvement in terms of performance for weather index insurance with respect to traditional approaches?

2. To which extent do the performances of machine learning models improve with the addition of input data?

3. Do the best performing models share similar properties? (e.g. use more input data or consistently have similar algorithm's features).

In this study we focus on the detection of two types of weather events with very different features: floods, which are mainly local events that can develop over a time scale going from few minutes to days, and droughts, which are creeping phenomena that involve widespread areas and have a slow onset and offset. In addition, floods cause immediate losses (Plate, 2002), while droughts produce non-structural damages and their effects are delayed with respect to the beginning of the event (Wilhite, 2000). Both satellite images and reanalysis are used as input data to show the potential of these instruments when properly designed and managed. Two of the most used machine learning methodologies, neural network (NN) and support vector machine (SVM), are applied. With ML models it is not always straightforward to know a priori which model(s) perform(s) better, or which model configuration(s) should be used. Therefore, various model configurations implemented starting from both SVM and NN are explored and a rigorous evaluation of their performances is accomplished. The best performing configurations are tested to reproduce past extreme events in a case study region.

Section 2 describes the NN and SVM algorithms used in this study and their configurations, the procedure adopted to take into consideration the problem of data imbalance due to the rarity of extreme events, the assessment of the quality of the predictions and the procedure used to select the best performing models and configurations. In addition, an overview of the used datasets is provided. Section 3 gives some insights on the area where the described methodology is applied. In section 4 the most important outcomes are shown for both floods and droughts.




## 2 Methodology

Machine learning is a subset of artificial intelligence whose main purpose is to give computers the possibility to learn, through-
out a training process, without being explicitly programmed (Samuel, 1959). It is possible to distinguish machine learning
models based on the kind of algorithm that they implement and the type of task that they are required to solve. Algorithms may
be divided into two broad groups: the ones using labelled data (Maini and Sabri, 2017), also known as supervised learning algo-
rithms, and the ones that during the training receive only input data for which the output variables are unknown (Ghahramani,
2004), also called unsupervised learning algorithms.

As previously mentioned, in index insurance, payouts are triggered whenever measurable indices exceed predefined thresh-
olds. From a machine learning perspective, this corresponds to an objective classification rule for predicting the occurrence of
loss or no loss based on the trigger variable. The rule can be developed using past training sets of hazard and loss data (su-
pervised learning). Conceptually, the development of a parametric trigger should correspond to an informed decision-making
process i.e. a process which, based on data, a-priori knowledge and an appropriate modelling framework, can lead to optimal
decisions and effective actions. This work aims to leverage the aptitude of machine learning, particularly supervised learning
algorithms, to support the decision-making process in the context of parametric risk transfer, applying NN and SVM for the
identification of extreme weather events, namely flooding and drought for this particular study.

Consider the occurrence of losses caused by a natural hazard on each time unit $t = 1, ..., T$ over a certain study area $G$, and
let $L_t$ be a binary variable defined as

$$L_t = \begin{cases} 0 & \text{if loss occurs on } t \text{ in } G \\ 1 & \text{if loss doesn't occurr on } t \text{ in } G \end{cases} \tag{1}$$

The aim is then to predict the occurrence of losses based on a set of explanatory variables obtained from non-linear transfor-
mations of a set of environmental variables. This hybrid approach aims to capture some of the physical processes of how the
hazard creates damage by incorporating a priori expert knowledge on environmental processes and damage-inducing mecha-
nisms for different hazards. Raw environmental variables are not always able to fully describe complex dynamics like flood
induced damage, therefore, the usage of expert knowledge is important to provide the machine learning model with input data
that are able to better characterize the natural hazard.

Supervised learning with machine learning methods based on physically-motivated transformations of environmental vari-
ables are then used to capture loss occurrence. The models are set up such that they produce probabilistic predictions of loss
rather than directly classifying events in a binary manner. This allows the parametric trigger to be optimized in a subsequent
step, in a metrics-based, objective and transparent manner, by disentangling the construction of the model from the decision-
making regarding the definition of the payout-triggering threshold. Probabilistic outputs are also able to provide informative
predictions of loss occurrence that convey uncertainty information, which can be useful for end users when a parametric model
is operational (Figueiredo et al., 2018).

Figure 1 summarizes the general framework implemented in this work.





**2.1    Variable and datasets selection**

The data-driven nature of ML models implies that the results yielded are as good as the data provided. Thus, the effectiveness of the methods depends heavily on the choice of the input variables, which should be able to represent the underlying physical process (Bowden et al., 2005). The data selection (and subsequent transformation) therefore requires a certain amount of a priori knowledge of the physical system under study. For the purpose of this work, precipitation and soil moisture were used

as input variables for both flood and drought. An excessive amount of rainfall is the initial trigger to any flood event (Barredo, 2007),while scarcity of precipitation is one of the main reasons that leads to drought periods (Tate and Gustard, 2000). Soil moisture is instead used as a descriptor of the condition of the soil. With the idea to implement a tool that can be exploited in the framework of parametric risk financing, we selected the datasets to retrieve the two variables according to five criteria:

1. Spatial resolution: a fine spatial resolution that takes into account the climatic features of the various areas of the con-
sidered country is needed to develop accurate parametric insurance products.

2. Frequency: the selected datasets should be able to match the duration of the extreme event that we need to identify. In the case of floods, which are quick phenomena, daily or hourly frequencies are required.

3. Spatial coverage: global spatial coverage enables the extension of the developed approach to areas different from the case study region.

4. Temporal coverage: since extreme events are rare, a temporal coverage of at least 20 years is necessary to allow a correct model calibration.

5. Latency time: a short latency time is necessary to develop tools capable of identifying extreme-events in near-real time.

Based on a comprehensive review of available datasets, we found six rainfall datasets and one soil moisture dataset matching the above criteria. With respect to the studies analysed in Mosavi et al. (2018), Hao et al. (2018) and Fung et al. (2019), that

associated a single dataset to each input variable, here six datasets are associated to a single variable (rainfall). The use of multiple datasets is able to improve the ability of models in identifying extreme events, as demonstrated for example by Chiang et al. (2007) in the case of flash floods. In addition, single datasets may not perform well; the combination of various datasets produces higher quality estimates (Chen et al., 2019). Two merged satellite-gauge products (the Climate Hazard Group Infrared precipitation, CHIRPS and the CPC Morphing technique, CMORPH,) and four satellite-only (the Global Satellite Mapping of

Precipitation GSMaP, the Integrated Multi-Satellite Retrievals for GPM, IMERG; the Precipitation Estimation from Remotely-Sensed Information using Artificial Neural Networks, PERSIANN; and the Global PERSIANN Cloud Classification System, PERSIANN-CCS) datasets were used. The main features of the selected datasets are reported in Table1.

Soil moisture was retrieved from the ERA5 reanalysis dataset, produced by the European Centre for Medium Range Weather Forecast (ECMWF). The dataset provides information on 4 soil moisture layers (Layer 1: 0-7 cm, Layer 2: 7-28cm; Layer 3:

28-100cm; Layer 4: 100-289cm). Table 2 shows the main features of the ERA5 dataset.


## 2.2 Data transformation

The raw environmental variables are subjected to a transformation which is dependent on the hazard at study and is deemed more appropriate to enhance the performances of the model, as described below.

### 2.2.1 Flood

Flood damage is not directly caused by rainfall, but from physical actions originated by water flowing and submerging assets usually located on land. As a result, even if floods are triggered by rainfall, a better predictor for the intensity of a flood and consequent occurrence of damage is warranted. To achieve this, we adopt a variable transformation to emulate the physical processes behind the occurrence of flood damage due to rainfall, based on the approach proposed by Figueiredo et al. (2018), which is now briefly described.

Let $X_t(g_j)$ epresent the rainfall amount accumulated over grid cell $g_j$ belonging $G$ on day $t$. Potential runoff is first estimated from daily rainfall. This corresponds to the amount of rainwater that is assumed to not infiltrate the soil and thus remain over the surface, and is given by

$$R_t(g_j) = max\{X_t(g_j) - u, 0\} \tag{2}$$

where $u$ is a constant parameter that represents the daily rate of infiltration.

Overland flow accumulates the excess of rainfall over the surface of a hydrological catchment. This process is modelled using a weighted moving time average, which preserves the accumulation effect and allows the contribution of rainfall on previous days to be considered. The moving average is restricted to a three-day period. The potential runoff volume accumulated over cell $g_j$ over days $t$, $t-1$, $t-2$ is thus given by

$$R_t^*(g_j) = \theta_0 R_t(g_j) + \theta_1 R_{t-1}(g_j) + \theta_2 R_{t-2}(g_j) \tag{3}$$

where $\theta_0, \theta_1, \theta_2 > 0$ and $\theta_0 + \theta_1 + \theta_2 = 1$

Finally, let $T_t$ be an explanatory variable representing potential flood intensity for day $t$, which is defined as

$$Y_t = \sum_{j=1}^{J} \frac{R_t^*(g_j)^\lambda - 1}{\lambda} \tag{4}$$

The Box-Cox transformation provides a flexible, non-linear approach to convert runoff to potential damage for each grid cell, which is summed over all grid cells in a study area to obtain a daily index of flood intensity. In order to obtain the $Y_t$

variable that best describes potential flood losses due to rainfall, the transformation parameters $u$, $\theta_1$, $\theta_2$ and $\lambda$ are optimized



by fitting a logistic regression model to concurrent potential flood intensity and loss data, and maximizing the likelihood using a quasi-Newton method:

$$L_t \sim Bernoulli(p_t) \tag{5}$$

with

$$log\left(\frac{p_t}{1-p_t}\right) = \beta_0 + \beta_1 Y_t \tag{6}$$

### 2.2.2 Drought

Before being processed by the ML model, rainfall data are used to compute the standardized precipitation index (SPI). The SPI is a commonly used drought index, proposed by Mckee et al. (1993). Based on a comparison between the long-term precipitation record (at a given location for a selected accumulation period) and the observed total precipitation amount (for

the same accumulation period), the SPI measures the precipitation anomaly. The long-term precipitation record is fitted to the gamma distribution function $(r(\alpha))$. The gamma distribution is defined according the following equation:

$$g(x) = \frac{1}{\beta^\alpha \Gamma(\alpha)} x^{\alpha-1} e^{-\frac{x}{\beta}} \, for \, x > 0 \tag{7}$$

where $\alpha$ and $\beta$ are respectively the shape factor and the scale factor. The two parameters are estimated using the maximum likelihood solutions according to the following equations:

$$\alpha = \frac{1 + \sqrt{1 + \frac{4D}{3}}}{4D} \tag{8}$$

$$\beta = \frac{\overline{x}}{\alpha} \tag{9}$$

where $\overline{x}$ is the mean of the distribution and $N$ is the number of observations.

The cumulative probability $G(x)$ is defined as

$$G(x) = \int_0^x g(x)dx = \frac{1}{\beta^\alpha \Gamma(\alpha)} \int_0^x x^{\alpha-1} e^{-\frac{x}{\beta}} dx \tag{10}$$

Since the gamma function is undefined if $x = 0$ and precipitation can be null, the definition of cumulative probability is adjusted to take into consideration the probability of a zero:

$$H(x) = q + (1-q)G(x) \tag{11}$$





where $q$ is the probability of a zero. $H$ is then transformed into the standard normal distribution to obtain the SPI value:

$$SPI = \phi^{-1}H(x) \tag{12}$$

where $\phi$ is the standard normal distribution.

The mean SPI value is therefore zero. Negative values indicate dry anomalies, while positive values indicate wet anomalies. Table 3 reports drought classification according to the SPI. Conventionally, drought starts when SPI is lower than -1. The drought event is ongoing until SPI is up to 0 (Mckee et al., 1993). The main strengths of the SPI stay in the fact that the index is standardized, therefore it can be used to compare different climate regimes, and that it can be computed for various

accumulation periods (World Meteorological Organization and Global Water Partnership, 2016). In this study SPI1, SPI3, SPI6 and SPI12 were computed. Shorter accumulation periods (1-3 months) are used to detect impacts on soil moisture and on agriculture. Medium accumulation periods (3-6 months) are preferred to identify reduced streamflow and longer accumulation periods (12-48 months) indicates reduced reservoir levels (European Drought Observatory, 2020).

### 2.3    Machine learning algorithms

We now focus on the machine learning algorithms adopted in this work, starting with a short introduction and description of their basic functioning, and next delving into the procedure used to build a large number of models based on the domain of possible configurations for each ML method. Finally, the metrics used to evaluate the models are introduced and the reasoning behind their selection is highlighted.

### 2.3.1    Neural Network (NN)

Neural networks drew inspiration from the behaviour of biological neurons in the human brain, where neurons interconnected by synapses are able to perform a function when activated (Vaiserman and Lushchak, 2018). In recent years and with the advent of big data, neural networks have been increasingly used to efficiently solve many real-world problems, related for example with pattern recognition and classification of satellite images (Dreyfus, 2005), where the capacity of this algorithm to handle nonlinearity can be put to fruition (Stevens and Antiga, 2019). Neural networks have three types of layers: the input layer, the

output layer, and one or more hidden layers between them. The number of neurons in the input layer is uniquely determined by the input data (e.g. number of environmental variables source), while the output layer, for binary classification problem, contains one hidden node returning a prediction about either of the two classes. Instead, the selection of the number of hidden layers and hidden nodes turns out to be more intricate and case-dependent. Although different methods have been presented to estimate the optimal neural network structure (Stathakis, 2009), there is no globally agreed-on procedure to derive the ideal

configuration of the network architecture (Mas and Flores, 2008). Depending on the number of layers, the neural network takes different names: artificial neural networks (ANN) are usually defined as networks with only one hidden layer; deep neural networks (DNN) are composed of two or more layers. The difference between ANN and DNN is not perfectly defined in literature, and for sake of simplicity, this paper will address them simply as neural networks, specifying where needed





the number of hidden layers and hidden nodes. Figure 2 displays the different parts composing a neural network and their interaction during the learning process. A neural network with multiple layers can be represented as a sequence of equations, where the output of a layer is the input of the following layer. Each equation is a linear transformation of the input data, multiplied by a weight ($w$) and the addition of a bias ($b$) to which a fixed nonlinear function is applied (also called activation function)

$$x_1 = f(w_0 x_0 + b_0) \tag{13}$$

$$x_2 = f(w_1 x_1 + b_1)$$

$$\vdots$$

$$y = f(w_n x_n + b_n)$$

The goal of these equations is to diminish the difference between the predicted output and the real output. This is attained by minimizing a so called loss function through the fine tuning of the parameters of the model, the weights. The latter procedure is carried out by an optimizer, whose job is to update the weights of the network based on the error returned by the loss function.

The iterative learning process can be summarized by the following steps:

1. Start the network with random weights and bias

2. Pass the input data and obtain a prediction

3. Compare the prediction with the real output and compute the loss function, which is the function that the learning process is trying to minimize.

4. Backpropagate the error, updating each parameter through an optimizer according to the loss function.

5. Iterate the previous step until the model is trained properly. This is achieved by stopping the training process when either the loss function is not decreasing anymore or when a monitored metric has stopped improving over a set amount of definition.

Specific to the training process, monitoring the training history can provide useful information, as this graphic representation of the process depicts the evolution over time of the loss function for both training and validation set. Looking at the history of the training has a twofold purpose: firstly, being the training a minimization problem, as long as the loss function is decreasing the model is still learning, while any eventual plateau or uprising would mean that the model is overfitting (or not learning anymore from the data). The latter is avoided when the loss function of training and validation dataset display the same decreasing trend (Stevens and Antiga, 2019). The monitoring assignment was exerted during the training of the model, where its capability to store the value of training and validation loss at each iteration of the process, enable the possibility to stop the training as soon as either losses are decreasing or plateauing over a certain amount of iterations. The neural network model


is created and trained using TensorFlow (Abadi et al., 2016). TensorFlow is an open-source machine learning library that was chosen for this work due to its flexibility, the capacity to exploit GPU cards to ease computational costs, its ability to represent a variety of algorithms and most importantly the possibility to carefully evaluate the training of the model.

### 2.3.2 Support Vector Machine (SVM)

Support vector machine is a supervised learning algorithm used mainly for classification analysis. They construct a hyperplane (or set of hyperplanes) defining a decision boundary between various data points representing observations in a multidimensional space. The aim is to create a hyperplane that separates the data on either side as homogeneously as possible. Among all possible hyperplanes, the one that creates the greatest separation between classes is selected. The support vectors are the points from each class that are the closest to the hyperplane (Wang, 2005). In parametric trigger modelling, as in many other real-world

applications, the relationships between variables are non-linear. A key feature of this technique is its ability to efficiently map the observations into a higher dimension space by using the so-called kernel trick. As a result, a non-linear relationship may be transformed into a linear one. A support vector machine can also be used to produce probabilistic predictions. This is achieved by using an appropriate method such as Platt scaling (Platt, 1999), which transforms its output into a probability distribution over classes by fitting a logistic regression model to a classifier's scores. The support vector machine algorithm was carried out

using the C-support vector classification (Boser et al., 1992) formulation implemented with the scikit-learn package in python (Pedregosa et al., 2011). Given training vectors $x_i \in R^p i = i, ..., l$ and a label vector $y \in \{0, 1\}^n$, this specific formulation is aimed at solving the following optimization problem:

$$min(w, b, \xi) = \frac{1}{2}\omega^t\omega + C\sum_{i=1}^{l}\xi_i \tag{14}$$

$subject\ to\ y_1(\omega^t\Psi(x_i) + b) \geq 1 - \xi_i$

$\xi_i \geq 0, i = 1, ..., l$

where $\omega$ and $b$ are adjustable parameters of the function generating the decision boundary, $\Psi_i$ is a function that projects $x_i$ into a higher dimensional space, $\xi_i$ is the slack variable and $C > 0$ is a regularization parameter, which regulates the margin

of the decision boundary allowing an increasing number of misclassification for lower value of $C$ and decreasing number of misclassification for higher C (Fig. 3).

### 2.3.3 Model construction

Hereinafter is proposed a procedure to assemble the machine learning models, that involves techniques borrowed from the data mining field and a deep understanding of all the components the algorithms are made of. The main purpose is to identify the

actions required to establish a robust chain of model construction. Figure 4, a zoom-in of the ML algorithm box of the previous workflow, describes the steps followed in order to create the best SVM and NN models, from the focus put on the importance of data enhancing to the selection of appropriate evaluation metrics keeping in mind the large variety of parameters comprising





these models and the wide ranges that these parameters can have. Hypothetically speaking, one may create a neural network with an infinite number of layers or a support vector machine model with infinite values of the $C$ regularization parameters.

*Pre-processing of data*

Data preprocessing (DPP) is a vital step to any ML undertaking, as the application of techniques aimed at improving the quality of the data before training leads to improvement of the accuracy of the models (Crone et al., 2006). Moreover, data preparation usually generates leaner and more reliable datasets, boosting the efficiency of the ML algorithm (Zhang et al., 2003). The literature presents several operations that can be adopted to transform the data depending on the type of task the

model is required to carry out (Huang et al., 2015; Felix and Lee, 2019). For instance, images and video analysis might require previous cropping or blurring through Gaussian convolution to better identify the edges of an image (Getreuer, 2013), while machine learning model used for time-series forecast benefit more from the detection of outliers and duplicate instances (Kotsiantis and Kanellopoulos, 2006). In this paper preprocessing operations were split into four categories: data quality assessment, data partitioning, feature scaling and resampling technique aimed at dealing with class imbalance. The first three

are crucial to application of any valid model, while the latter is required when dealing with the classification of rare events. Data quality assessment was carried out to ensure the validity of the input data, filtering out any anomalous value (e.g. negative values of rainfall). Also, this process was used to identify any incoherence amid the dataset for example by checking the spatial patterns of precipitation in the days leading to flood events.

The partitioning of the dataset into training, validation and testing portions is fundamental to give the model the ability to

learn from the data and avoid a problem often encountered in ML application: overfitting. This phenomenon takes place when a model starts overlearning from the training dataset, picking up patterns that belong solely to the specific set of data it is training on and that are not depictive of the real-world application at hand, making the model unable to generalize to sample outside this specific set of data. To avoid overfitting one should split the data into at least 2 parts (McClure, 2017). The training set, upon which the model will learn, and a validation dataset functioning as a counterpart during the training process of the model,

where the losses obtained from the training set and those obtained from the validation set are compared to avoid overfitting. A further step would be to set aside a testing set of data that the model has never seen. Feeding data that the model has never encountered before is an excellent indicator of the model ability to generalize. The splitting of the data is key to the validation of the model. In this work, a k-fold cross validation (Mosteller and Tukey, 1968) was used to validate the SVM model, whilst for NN, TensorFlow allows the user to declare a percentage of the data that is retained as validation data at each iteration of the

training loop, therefore, embedding the validation process into the construction of the model.

Feature scaling is a procedure aimed at improving the quality of the data by scaling and normalizing numeric values so as to help the ML model in handling varying data in magnitude or unit (Aksoy and Haralick, 2001). The variables could be rescaled to the $[0,1]$ range or to the $[-1,1]$ range or normalized subtracting the mean and dividing by the standard deviation. The scaling is carried on after the splitting of the data and is usually calibrated over the training data, and then, the testing set is

scaled with the mean and variance of the training variables (Mueller and Massaron). In classification problems, class imbalance usually reflects an unequal distribution of classes within a dataset. Imbalance means that the number of data points available for different classes is significantly different; if there are two classes, a balanced dataset would have approximately 50% points



for each of the classes. For most machine learning techniques, little imbalance is not a problem, but when the class imbalance
is high, e.g. 85% points for one class and 15% for the other, standard optimization criteria or performance measures may not
be as effective as expected (Garcia et al., 2012). Extreme events are by definition rare, hence, the imbalance existing in the
dataset should be addressed. One approach to address imbalances is using resampling techniques such as over-sampling (Ling
and Li, 1998) and SMOTE (Chawla et al., 2002). Over-sampling is the process of up-sampling the minority class by randomly
duplicating its elements. SMOTE (Synthetic Minority Over-sampling Technique) involves the synthetic generation of data
looking at the feature space for the minority class data points and considering its $k$ nearest neighbour where $k$ is the desired
number of synthetic generated data. Another possible approach to address imbalances is weight balancing, which restores
balance in the data by altering the way the model "looks" at the under-represented class. Both scikit-learn and TensorFlow
allow for the implementation of class weight into the model construction through an explicit parameter. The weighting values
can easily be tweaked to find the optimal settings for a given problem.

*Analysis of model configurations*

Up to this point, several models characteristics and a considerable amount of possible operations aimed at data augmentation
were presented creating an almost boundless domain of model configurations. In order to explore such domain, for each ML
method multiple key aspects were tested. Both methods shared an initial investigation of the sampling technique and the
combination of input dataset that shall be fed into the models; all the data augmentation techniques previously introduced were
tested along with the data in their pristine condition where the model tries to overcome the class imbalance by himself. All
the possible combinations of input dataset were tested starting from one dataset for SVM and with two dataset for NN up to
the maximum number of environmental variables used. The latter procedure can be used to determine whether the addition of
new information is beneficial to the predictive skill of the model and also to identify which dataset provides the most relevant
information.

As previously discussed, these models present a multitude of customizable facets and parameters. For support vector ma-
chine, the regularization parameter C and the kernel type were the elements chosen as the changing parts of the algorithm. Five
different values of C were adopted, starting from a soft margin of the decision boundary moving towards narrower margins,
while three kinds of kernel functions were used to find the separating hyperplane: linear, polynomial and radial. The setup for a
neural network is more complex and requires the involvement of more parameters, namely, the loss function and the optimizers
concerning the training process, plus, the number of layers and nodes and the activation functions as key building blocks of
the model architecture. Each of the aforementioned parameters can be chosen among a wide range of options; moreover, there
is not a clear indication for the number of hidden layers or hidden nodes that should be used for a given problem. Thus, for
the purpose of this study, the intention was to start from what was deemed the "standard" for the classification task for each
of these parameters, deviating from these standard criteria towards more niche instances of the parameters trying to cover as
much as possible of the entire domain.




**2.4 Evaluation of predictive performance**

The evaluation of the predictive performance of the models is fundamental to select the best configuration inside the entire realm of possible configurations. A reliable tool to objectively measure the differences between model outputs and observations is the confusion matrix. Table 4 shows a schematic confusion matrix for a binary classification case. When dealing with thousands of configurations and, for each configuration, with an associated range of possible threshold probabilities, it is impracticable

to manually check a table or a graph for each setup of the model. Therefore, a numeric value, also called evaluation metric, is often employed to synthetize the information provided by the confusion matrix and describe the capability of a model (e.g. (M and M.N, 2015)).

There are basic measures that are obtained from the predictions of the model for a single threshold value. These include the precision, sensitivity, specificity and false alarm rate, which take into consideration only one row or column of the confusion

matrix, thus overlooking other elements of the matrix (e.g. high precision may be achieved by a model that is predicting a high value of false negatives). Nonetheless, they are staples in the evaluation of binary classification, providing insightful information depending on the problem addressed. Accuracy and F1 score, on the other hand, are obtained by considering both directions of the confusion matrix, thus giving a score that incorporates both correct predictions and misclassifications. The accuracy is the ratio between the correct prediction over all the instances of the dataset, and is able to tell how often, overall, a

model is correct. The F1 score is the harmonic mean of precision and recall. In its general formulation derived from Jones and Van Rijsbergen (1976)'s effectiveness measure, one may define a $F_\beta$ score for any positive real $\beta$ (Eq. 2.4):

$$F_\beta = 1 + \beta^2 \frac{precision * sensitivity}{(\beta^2 * precision) + sensitivity} \tag{15}$$

where $\beta$ denotes the importance assigned to precision and sensitivity. In the F1 Score both are considered to have the same

weight. For values of $\beta$ higher than one more significance is given to false negatives, while $\beta$ lower than one puts attention on the false positive.

The goodness of a model may also be assessed in broader terms with the aid of Receiver Operating Characteristic (ROC) and Precision-Sensitivity curves (PS). The ROC curve is widely employed and is obtained plotting the sensitivity against the false alarm rate over the range of possible trigger thresholds (Krzanowski and Hand, 2009). The PS curve, as the name suggests, is

obtained plotting the precision against the sensitivity over the range offor a changing thresholds. For this work, the threshold corresponds to the range of probabilities between 0 and 1. These methods allow evaluating a model in terms of its overall performance over the range of probabilities, by calculating the so-called area under curve (AUC).

It should be noted that both ROC curve and the accuracy metric should be used with caution when class imbalance is involved (Saito and Rehmsmeier, 2015), as having a large amount of true negative tends to result in low value of FPR (or 1-

specificity). Also, while here we focus on performance-based evaluation measures, an alternative approach may be to quantify the utility of the predictive systems. By taking into account actual user expenses and thus specific weights for different model outcomes, a utility-based approach may potentially lead to different decisions regarding model selection and definition of the





trigger threshold (Murphy and Ehrendorfer, 1987; Figueiredo et al., 2018). This aspect is outside the scope of the present article and warrants further research.

Table 5 summarizes the metrics described above used in this paper to evaluate model performances.

  In the context of performance evaluation, it is also relevant to discuss the issue of class imbalance. Class imbalance refers to the difference between positive and negative instances with the latter usually outnumbering the former. Thus, it is important to keep in mind how class imbalance might affect measures that use the true negative in their computation. Saito and Rehmsmeier (2015) tested several metrics on datasets with varying class imbalance, and showed how accuracy, sensitivity and specificity

are insensitive to the class imbalance. This kind of behaviour from a metric can be dangerous and definitely misleading when assessing the performances of a ML algorithm and might lead to the selection of a poorly designed model (Sun et al., 2009). Lastly, once the domain of all configurations is well established and the best settings of the ML algorithms are extracted from it through the aforementioned metrics, the predictive performances of the models are compared to those of logistic regression (LR) models. The logistic regression is a more traditional statistical model whose application to index insurance has recently

been proposed, and can be said to already represent in itself an improvement over common practice in the field (Calvet et al., 2017; Figueiredo et al., 2018). Thus, this comparison is able to provide an idea about the overall advantages of using a ML method.

## 3 Case study

This study adopts the Dominican Republic as its case study. The Dominican Republic is located on the eastern part of the

island of Hispaniola, one of the Greater Antilles, in the Caribbean region. Its area is approximatively $48,671 km^2$. The central and western parts of the county are mountainous, while extensive lowlands dominate the southeast (Izzo et al., 2010).

  The climate of the Dominican Republic is classified as "tropical rainforest". However, due to its topography, the country's climate shows considerable variations over short distances. The average annual temperature is about 25 °C, with January being the coldest month (average monthly temperature over the period 1901-2009 of about 22 °C) and August the hottest (average

monthly temperature over the period 1901-2009 of about 26 °C) (World Bank, 2019). Rainfall varies from 700 to 2400 mm per year, depending on the region (Payano-Almanzar and Rodriguez, 2018). The six considered rainfall datasets (described in 1) exhibit huge differences in average annual precipitation values (Fig. 5). CMORPH shows the lowest values, CHIRPS and IMERG the highest ones. The difference among absolute precipitation values does not affect the results of this study since precipitation is transformed into potential damage or SPI, as described in section 2.2.2 and therefore only relative values are

considered. It's interesting to note that all the datasets show similar precipitation patterns; on average, over the period from 2000 to 2019, rainfall was mainly concentrated in north-western regions, along the Haitian borders, with the south-western regions being the driest. The situation is different when considering the average soil moisture (Fig. 6). The central regions are the wettest, while the driest areas are located on the coast. There are no significant differences among the four soil moisture layers.





Weather-related disasters have a significant impact on the economy of the Dominican Republic. The country is ranked as the 10th most vulnerable in the world and the second in the Caribbean, as per the Climate Risk Index for 1997-2016 report (Eckstein et al., 2017). It has been affected by spatial and temporal changes in precipitation, sea level rise, and increased intensity and frequency of extreme weather events. Climate events such as droughts and floods have had significant impacts on all the sectors of the country's economy, resulting in socio-economic consequences and food insecurity for the country.

Over the period from 1960 to present, the most frequent natural disasters were tropical cyclones (45% of the total natural disasters that hit the country), followed by floods (37%) according to the International Disaster Database EMDAT (CRED, 2019). Floods, storms and droughts were the disasters that affected the largest number of people and caused huge economic losses.

        The accurate identification of past events is a key aspect in obtaining good results from the ML algorithm. A careful detection
of reported events, even if it is a time-consuming task, is essential to bridge together the a priori knowledge with the ability of ML models to exploit the data. Therefore, a wide range of text-based sources have been consulted to retrieve information on past floods and droughts that hit the Dominican Republic over the period from 2000 to 2019. International disasters databases, such as the world renowned EMDAT, Desinventar and ReliefWeb have been considered as primary sources. The events reported by the datasets have been compared with the ones present in hazard-specific datasets (such as FloodList and the Dartmouth
Flood Observatory) and in specific literature (Payano-Almanzar and Rodriguez, 2018; Herrera and Ault, 2017) to produce a reliable catalogue of historical events. Only events reported by more than one source were included in the catalogue. Figure 7 shows the past floods and droughts hitting the Dominican Republic over the period from 2000 to 2019. More details on the events can be found in Table A1 (floods) and A2 (droughts).

## 4   Results and Discussion

The results are presented in this section separating the two types of extreme events investigated, flood and drought. As described in section 2, both SVM and NN models require the assembling of several components. Table 6 collects the number of model configurations explored, broken down by type of hazard and ML algorithm with their respective parameters. The main differences between the ML models parameters for the two hazards reside in which data are provided to the algorithm and which sampling techniques are adopted. The input dataset combination were chosen as follows:

1. All the possible combinations from 1 up to 6 rainfall datasets (for neural network 2 rainfall datasets were considered the starting point).

        2. The remaining combinations are obtained adding progressively layers of soil moisture to the ensemble of six rainfall datasets.

        3. The drought case required the investigation of the SPI over different accumulation periods. One, three, six and twelve
455         months SPI were used.





Support vector machines and neural networks alike are able to return predictions (i.e. outputs) as a probability when the activation function allows it (e.g. sigmoid function), enabling the possibility to find an optimal value of probability to assess the quality of the predictions . Therefore, for each hazard, the results are presented by introducing at first the models achieving the highest value of the F1 score for a given configuration and threshold probability (i.e. a point in the ROC or precision-
sensitivity space). Secondly, the best performing model configurations for the whole range of probabilities according to the AUC of the precision-sensitivity curve are presented and discussed. The reasoning behind the selection of these metrics is discussed previously, in section 2.4. As described in the same section, the performances of the ML algorithms are evaluated through a comparison with a LR model.

### 4.1   Flooding

The flood case presented a strong challenge from the data point of view. Inspecting the historical catalogue of events the case study reported 5516 days with no flood events occurring and 156 days of flood, meaning approximately a $35 : 1$ ratio of no event/event. This strong imbalance required the use of the data augmentation techniques presented in section 2.3.1. The highest F1 score for the support vector machine was attained by the model configuration using an unweighted model taking advantage of all ten environmental variables with radial basis function as kernel type and a $C$ parameter equal to 500 (i.e. harder margin).
The neural network settings returning the highest F1 score were once again given by the model using all ten datasets applying an over-sampling to the input data. The network architecture was made up of 9 hidden layers with the amount of nodes for each layer as already described, activated by a ReLu function. The loss function adopted was the binary cross entropy and the weight update were regulated by an Adam optimizer. The evaluation metrics in the table refer to results measured on the testing set, therefore, never seen by the model. Overall, the two ML methods outperform decisively the logistic regression with
a slightly higher F1 score for the neural network.

In Fig. 8, panel (a) the highest F1 scores by method are reported in the precision-sensitivity space along with all the points belonging to the top 1% configurations according to F1 score. The separation is evident between the ML methods and the logistic regression, also, the plot highlights denser cloud of orange points in the upper left corner and denser cloud of red points in the lower right corner attesting, on average, an higher precision achieved by the SVMs and an higher sensitivity by
the NNs.

Figure 8, panel (b), depicts the goodness of SVM and NN versus the LR model, showing how the F1 scores of the best-performing settings for each of the three methods vary by increasing the number of input datasets. This plot shows that the SVM and LR models have similar performances up to the second layer of soil moisture, while NN performs considerably better overall. The NN and the SVM as opposed to the LR, show an increase in the performances of the models with increasing
information provided. The LR seems to plateau after 4 rainfall dataset and the improvements are minimal after the first layer of soil moisture is fed to the model. This would suggest, as expected, that the ML algorithms are better equipped to treat larger amounts of data.

Figure 9 presents the best-performing configurations according to the area under the PS curve. For neural network, this configuration is the one that also contains the highest F1 score, whereas the support vector machine the optimal configuration




shares the same feature of the one with the best F1 score with the exception of a softer decision boundary in the form of $C$ equal to 100. The results reported in Fig. 9 (a) and (b) about the best-performing configurations are further confirmation of the importance of picking the right compound measurement to evaluate the predictive skill of a model. In fact, according to the metrics using the true negative in their computation (i.e. specificity, accuracy and ROC) , one may think that these models are rather good, and this deceitful behaviour is not scaled appropriately for very bad models. The aim of this work is to correctly

identify a flood event rather than being correct when none occur, hence, overlooking the correct rejections seems reasonable.

Panel (a) and (b) of Fig. 9 shed a light on the inaccuracy of the ROC curve and the relative area under the curve (AUC). On the left are displayed the ROC curves, whilst on the right the PS curves of the ideal configurations for each method according to the highest AUC. The points in both curves represent a 0.01 increment in the trigger probability. The receiving operator curve indicates the NN as the worst model being the closest to the 45°line and having, along with SVM, a lower AUC with respect to

the logistic model. This signal is strongly contradicted by other metrics and the precision-sensitivity curve, where the red dots are the closest to the upper-right corner where the perfect model resides. The behaviour of these curves is linked, once again, to the disparity in the classes. Additionally, looking at panel (b), all models are pretty distant from the always-positive classifier (black hyperbole) more appropriate as a baseline to beat than a random classifier (Flach and Kull, 2015).

Panel (c) shows the behaviour of the prediction return by the ML models over the whole range of probabilities. It is noticeable

that although the peak value of F1 score is very close for both ML methods, the neural network displays steadier prediction over an extended range of probabilities. In fact, a robust identification of the true positive and low variability in false positive and false negative detection allows the model to have strong performances independently of which probability threshold one may choose.

Figure 10 portrays the properties of the top one-percent model configuration for both methods according to the area under

the PS curve. Support vector machine algorithms use the highest value of the C-parameter, which is the one used by the configuration attaining the highest F1 score. A bigger divide can be observed amid the sampling technique and the kernel function, where data input with no manipulation provided (i.e. Unw) is the most recurrent option occurring more than 40% of the time; similar percentage is attained by the radial basis function. On the other hand, neural networks prefer the adoption of oversampling to enhance the input data and almost 60% of the configurations use a rectified linear unit function to activate its

layer. Relative to the architecture of the network, a double peak can be observed at 8 and 9 layers, where the best-performing configurations can be found but it is noticeable an even larger presence of model configuration with 3 and 4 layers.

## 4.2  Drought

The data transformation for drought required the computation of the SPI from the precipitation data. The SPI was computed for different accumulation periods: Shorter accumulation periods (1-3 months) detect immediate impacts of drought (on soil

moisture and on agriculture), while longer accumulation periods (12 months) indicate reduced streamflow and reservoir levels. As shown in tables B1 and B2 models using SPI6 and SPI12 showed the best results and the values of the metrics are close to each other, thus, for brevity and in favor of clarity only one of the two is reported, namely, SPI over a six month accumulation period. Contrary to the flood case, the drought historical catalogue of events reported 1283 weeks with no droughts and 696




weeks of drought, with a ratio around $1.85 : 1$ of no event/event. Albeit balanced, models with weights assigned were also
investigated. The performances of the neural networks and the Support Vector were evaluated, like before, by a mix of metrics
and curves and a comparison against a logistic regression. It is important to point out that SPI is updated at weekly scale, same
temporal resolution of the predictions implying that each week counts as an event. This strong limitation is an important aspect
to keep in mind when analysing the results obtained for these models. The highest F1 scores for the drought case were obtained
from the unweighted model using all ten environmental variables with radial basis function as kernel type and a $C$ parameter
equal to 100 for SVM. The neural network model was at its best using all the datasets with weights for the classes. The network
architecture was made up of 8 hidden layers with the relative amount of nodes, activated by a ReLu function. The loss function
adopted was the binary cross entropy and the parameters update were regulated by an Adam optimizer.

Table 8 reports the metrics for the three methods, where the ML algorithms are a strong improvement with due respect to
the logistic regression, showing high value across all the prediction skill measurement. The SVM results as the more precise
model, while the NN is the most accurate overall. The implementation of either model should take into account the job that
these models are required to take on. A task that would require a stronger focus on the minimization of the false positive
(reduce the number of false alarm) one should elect to use the SVM, on the other hand, if the purpose of the model is to
balance the issue of false alarm and missed event one should veer towards the NN. Figure 11 remarks the distance between the
ML method and the logistics regression as well as echo what already observed for flood that the points for SVM tend to stay
on the precision side of the plot while the NN points gravitate towards the area of the plot with higher sensitivity value.

The addition of further dataset is still beneficial to the performances of the ML methods as displayed by Fig. 11, panel (b).
The increasing trend for both ML models start to slow down from the fourth rainfall datasets onward, which might be due to
the redundancy of the rainfall datasets. On the other hand, the addition of the layers of soil moisture improves the performances
especially for the support vector machine, which keeps improving steadily reaching the highest value of F1 Score when the
whole set of information is fed to the model.

Figure 12 refers to the best-performing configurations identified as the one with the highest area under the precision-
sensitivity curve. The best configurations for either neural network and support vector machine are the one containing the
point with the highest F1 score, thus having the same features previously listed. The disparities between classes for drought are
closer than those for flood, in turns giving the accuracy, and the ROC curve, more reliability from a quality assessment point
of view. Looking at the panels of fig. 12, both ROC and precision-sensitivity curves show the ML methods decisively outper-
forming the no-skill and always-positive classifier. Furthermore, both plots exhibit a tendency of the neural network to group
the points closer to each other towards the area containing the ideal model, which may indicate a more dependable prediction
of the events as indicated by panel (c). In fact, while the two configurations have a high value of F1 score for a wide range of
probabilities, the neural network has steadier prediction of true positive, false positive and false negative. This behaviour of the
neural network could also be linked to the miscalibration of the confidence (i.e. distance between the probability returned by
the model and the ground truth) associated with the predicted probability (Guo et al., 2017). The phenomenon arose with the
advent of modern neural networks that employing several layers (i.e. tens and hundreds) and a multitude of nodes were able
to improve the accuracy of their prediction while worsening the confidence of said prediction. Indeed, a miscalibrated neural





network would return a probability that would not reflect the likelihood that the event will occur turning into a numeric output
produced by the model.

The features breakdown of the model configurations top one-percent shown in Fig. 13, denotes a marked component of
the models using harder margins (i.e. high values of C-par) and radial basis function as a kernel for SVM. The best NNs
configurations are predominantly the ones using weight for the two classes, and the ReLu activation function. Also, a large
number of models use a high number of layers in accordance with the configuration with the highest area under the PS curve.
The fact that most of the configurations obtaining the best performances have deeper layers may be a confirmation of the
miscalibration affecting the estimated probabilities.

## 5    Conclusions

In this study we developed and implemented a machine learning framework with the aim of improving the identification of
extreme events, particularly for parametric insurance. The framework merges a priori knowledge of the underlying physical
processes of weather events with the ability of ML methods to efficiently exploit big data, and can be used to support informed
decision making regarding the selection of a model and the definition of a trigger threshold. Support vector machine and
neural network models were used to classify flood and drought events for the Dominican Republic, using satellite data of
environmental variables describing these two types of natural hazard. Model performance was assessed using state of the art
evaluation metrics. In this context we also discussed the importance of using appropriate metrics to evaluate the performances
of the models, especially when dealing with extreme events that may have a strong influence on some performance evaluators.

The proposed approach involves a preceding data manipulation phase where the data are preprocessed to enhance the per-
formances of the ML methods. A procedure aimed at designing and selecting the best parameters for the models was also
introduced. Once trained, the ML algorithms decisively outperformed the logistic regression, here used as a baseline for both
hazards. The predictive skill of both SVM and NN improved with increasing information fed to the models; indeed, the best
performances were always obtained by models using the maximum amount of data available , hinting at the possibility of in-
troducing additional and more diverse environmental variables to further improve the results. While the ML models performed
well for both hazards, the drought case showed exceptionally high values for all the adopted model evaluation metrics. This
discrepancy in the results between flood and drought might have several explanations. Indeed, the two hazards behave differ-
ently both in time and space. On one side, the aggregation at national scale is surely an obstacle for a rather local phenomenon
like flood. On the other side, defining a drought event weekly could be misleading since droughts are events spanning several
months, even years. Going at a higher resolution (e.g. regional scale) and introducing data describing the terrain of the area
should enhance the detection of flood events. For the drought case, introducing a threshold for the number of consecutive weeks
predicted before considering an event, or contemplating weekly predictions as a fraction of the overall duration of the event
are extensions to this work that deserve investigation to address the issue of potential overestimation of predictive skill.
Neural networks showed more robustness when compared to support vector machines, showing a higher value of F1 score
for a wide range of parameters. As already mentioned, this insensitiveness of NN to the probability threshold adopted may be





reconducted to the inability of the model to reproduce probability estimates that are a fair representation of the likelihood of occurrence of the event. Further developments of neural network models should take into consideration procedures that allow the assessment and the quantification of the confidence calibration of probability estimates.

A preliminary investigation of the characteristics shared by the best-performing model showed that some features are more relevant than others when building the ML model, depending on the type of algorithm and also the type of hazard. An in-depth study of how the performances of the models change when changing model properties could highlight which are the most important properties of the model to tune, speeding up the model construction phase and reducing the computational cost of running the algorithms.

Although several issues raised in this article warrant further research, there is clear potential in the application of machine algorithms to take advantage of increasing amounts of available environmental data within the context of weather index insurance. The framework presented and topics discussed in this study provide a scientific basis for the development of robust and operationalizable parametric risk transfer products.

*Data availability.* The six rainfall datasets (CCS, CHIRPS, CMORPH, GSMaP, IMERG and PERSIANN) and the soil mosture dataset
(ERA5) are freely available at the links cited in the references.

**Appendix A: Catalogue of historical events**

The following tables report the catalogue of historical events for floods (Table A1) and droughts (Table A2).

**Appendix B: Performance of NN and SVM in drought events identification when using different SPI accumulation periods**

The following tables report the performances of NN and SVM in drought events identification when using different SPI accumulation periods.

*Author contributions.* All authors contributed to the conceptual design of this study. LC and RF designed the modelling framework. LC implemented the models and ran the analyses. All authors analysed and discussed the results. LC, RF and BM wrote the manuscript with the support of MLVM. All authors reviewed the manuscript before submission to the journal.

*Competing interests.* The authors declare that they have no conflict of interest.





*Acknowledgements.* The research leading to these results has received funding from the Disaster Risk Financing Challenge Fund of the World Bank Group in the context of the SMART (A Statistical Machine Learning Framework for Parametric Risk Transfer) project. The research has been developed within the framework of the project 'Dipartimenti di Eccellenza', funded by the Italian Ministry of Education, University and Research at IUSS Pavia.



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

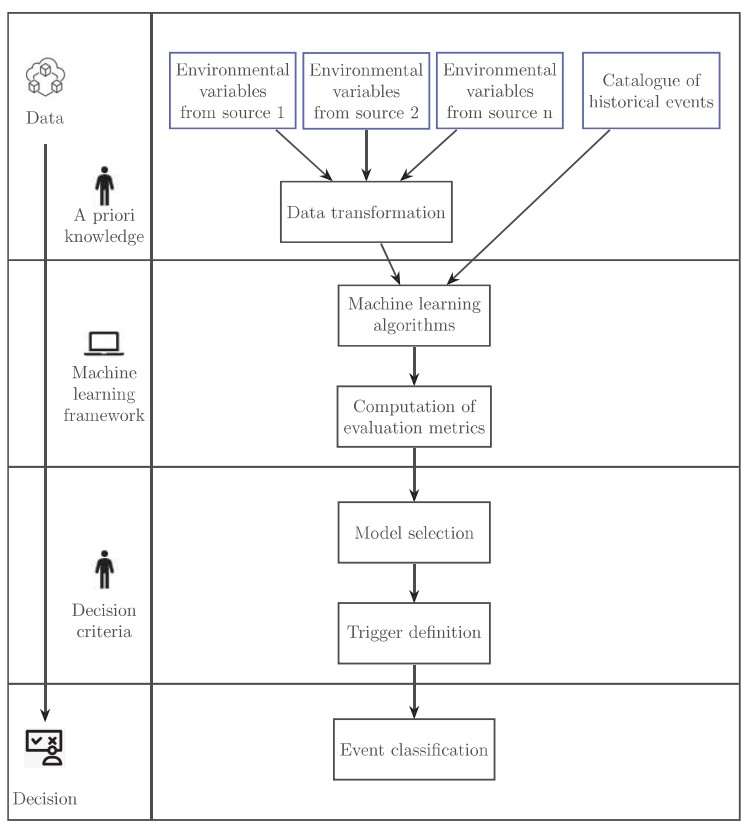

**Figure 1.** Flowchart of the proposed approach.





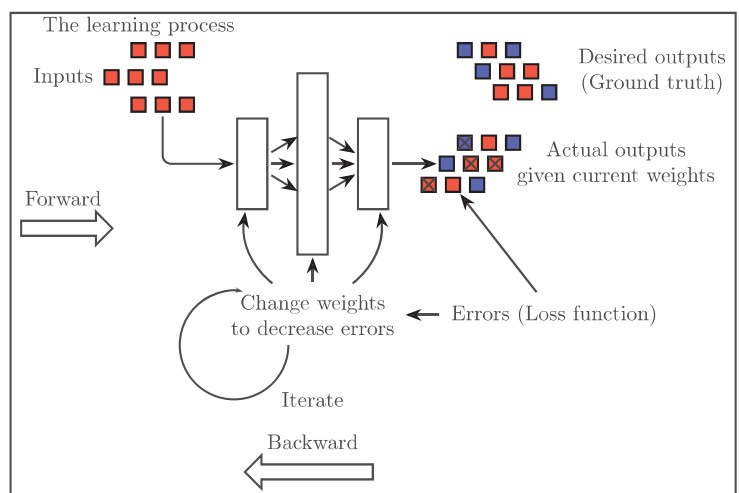

**Figure 2.** Learning process of a neural network (Stevens and Antiga, 2019).



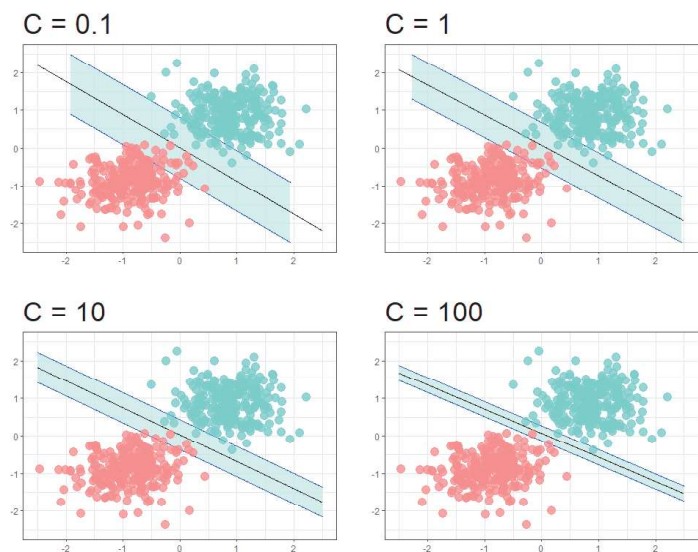

**Figure 3.** Decision boundary of support vector machine's algorithm, with changing regularization parameter C.

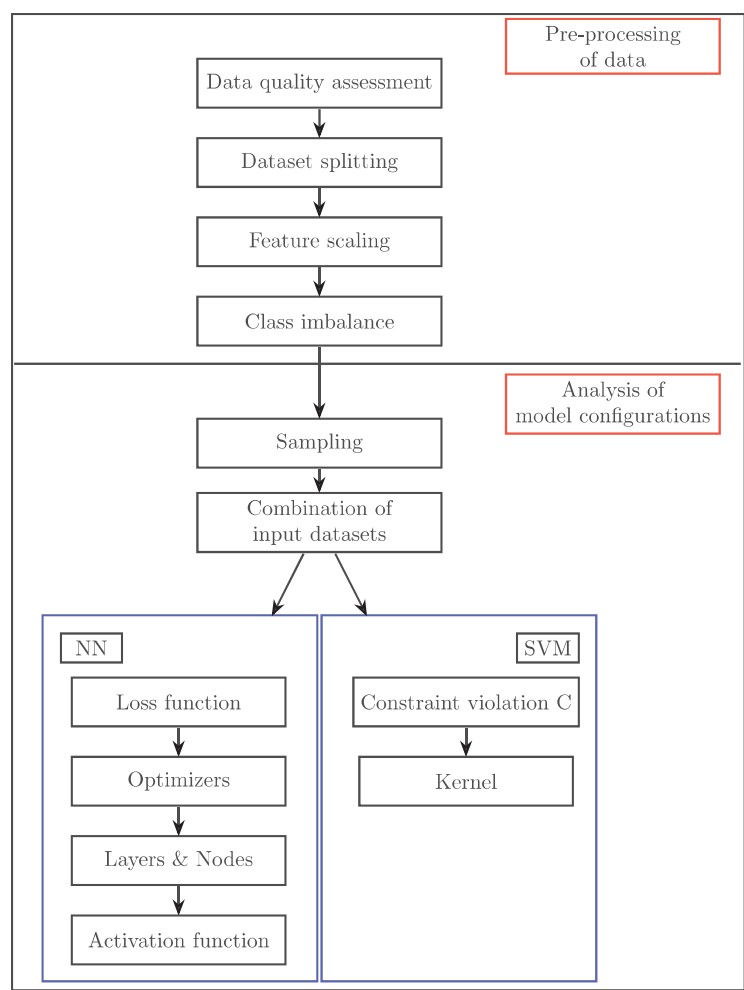

**Figure 4.** Framework used to analyse the domain of possible model configurations.



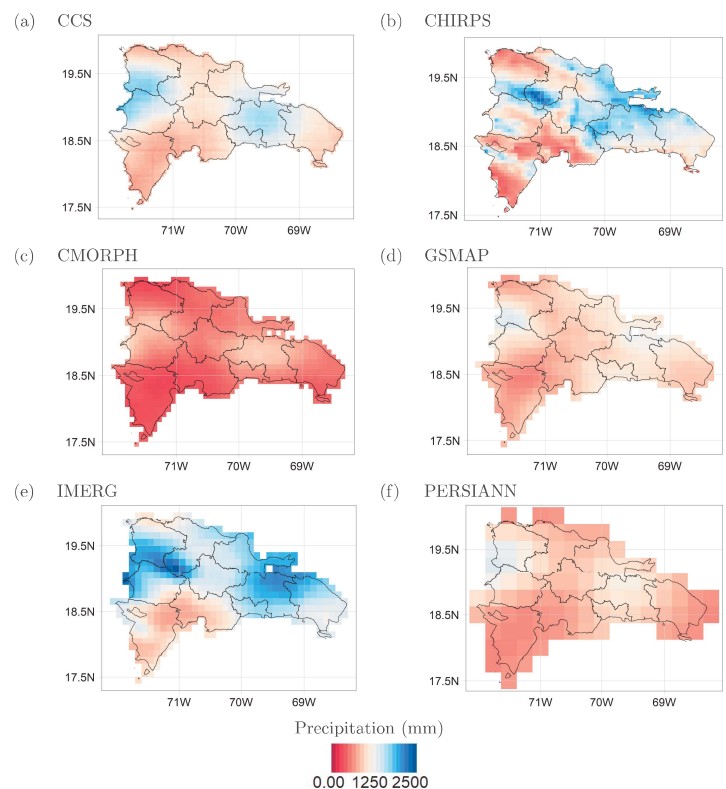

**Figure 5.** Average annual rainfall over the Dominican Republic according to the six considered datasets. (a) CCS, (b) CHIRPS, (c) CMORPH, (d) IMERG, (e) GSMaP, (f) PERSIANN.


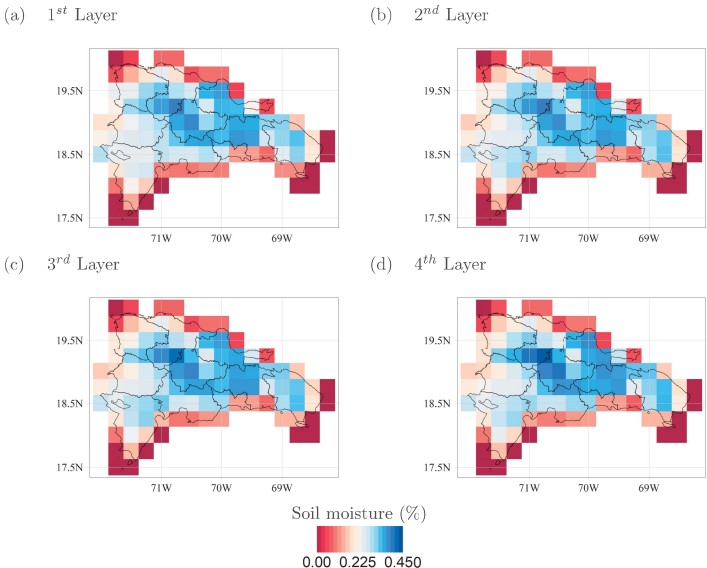

**Figure 6.** Average soil moisture over the Dominican Republic in the four soil moisture layers. (a) First layer, 0-7 cm, (b) Second layer, 7-28 cm, (c) Third layer, 28-100 cm, (d) Fourth layer, 100-289 cm.





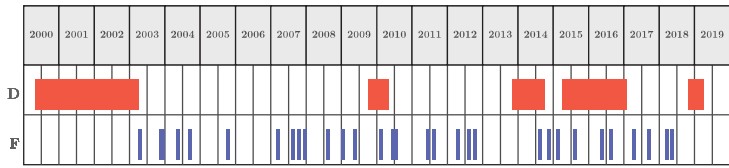

**Figure 7.** Overview of floods and droughts hitting the Dominican Republic over the period 2000-2019.

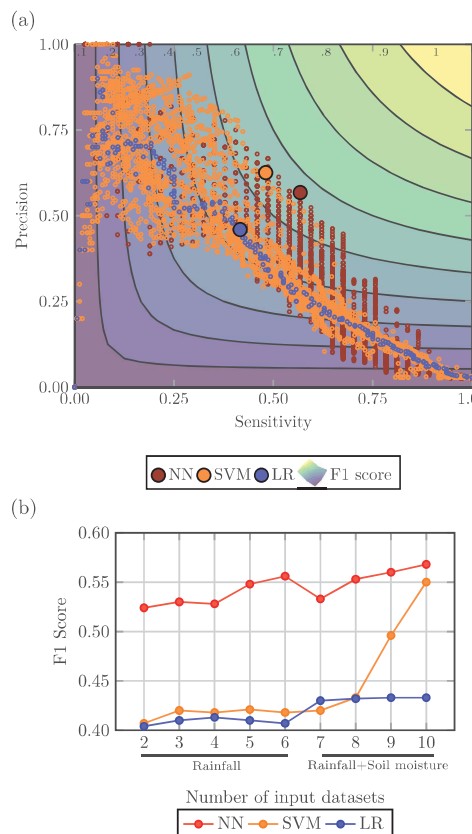

**Figure 8.** (a): Performance evaluation for the flood case: (a) Performances of the top 1% configurations in the precision-sensitivity space highlighting the highest F1 score, (b): Comparison of ML methods with LR with combination using increasing number of input datasets.


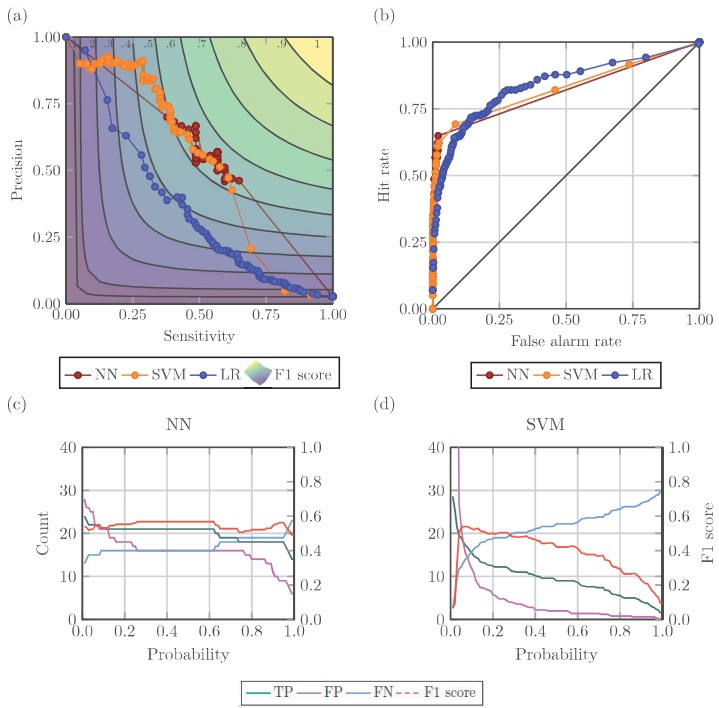

**Figure 9.** Best-performing configurations for the flood case: (a) PS curve, (b) ROC curve, (c) Variation of true positive, false positive,true negative and F1 score for the range of probability in NN and (d) in SVM.

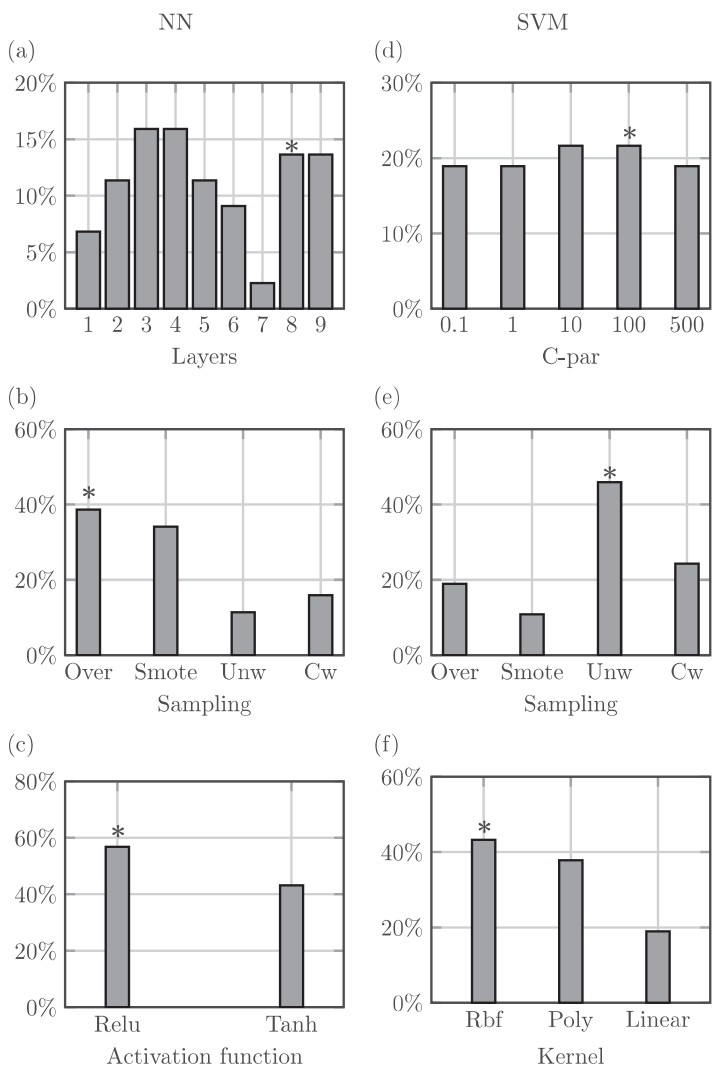

**Figure 10.** Properties of top 1% model configurations for the flood case. The stars denote the characteristics of the best-performing configu-rations according to the highest area under the precision-sensitivity curve.


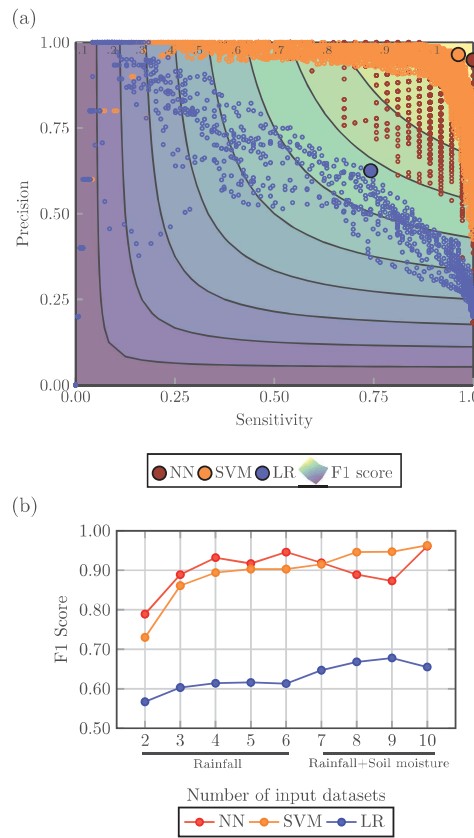

**Figure 11.** Performance evaluation for the drought case: (a) Performances of the top 1% configurations in the precision-sensitivity space highlighting the highest F1 score, (b): Comparison of ML methods with LR with combination using increasing number of input datasets.


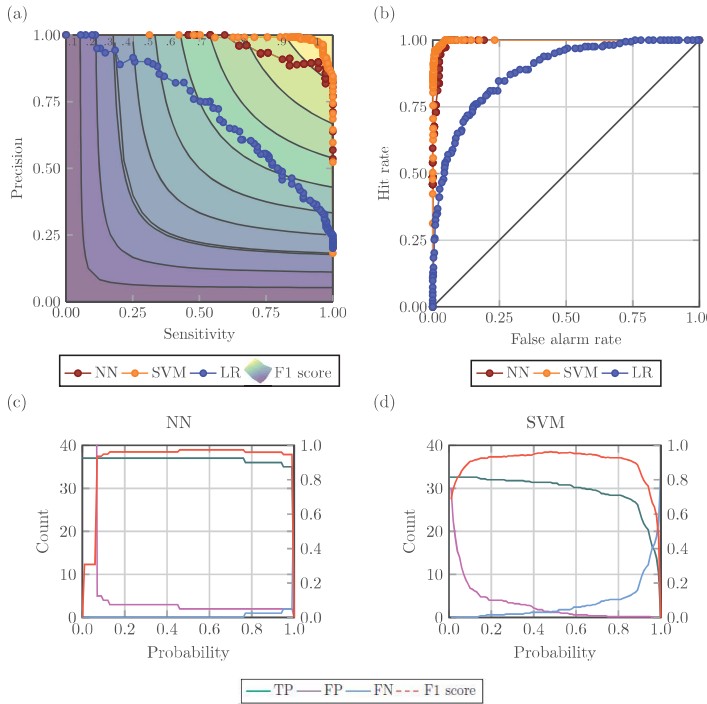

**Figure 12.** Best-performing configurations for the drought case: (a) PS curve, (b) ROC curve, (c) Variation of true positive, false positive,true negative and F1 score for the range of probability in NN and (d) in SVM.

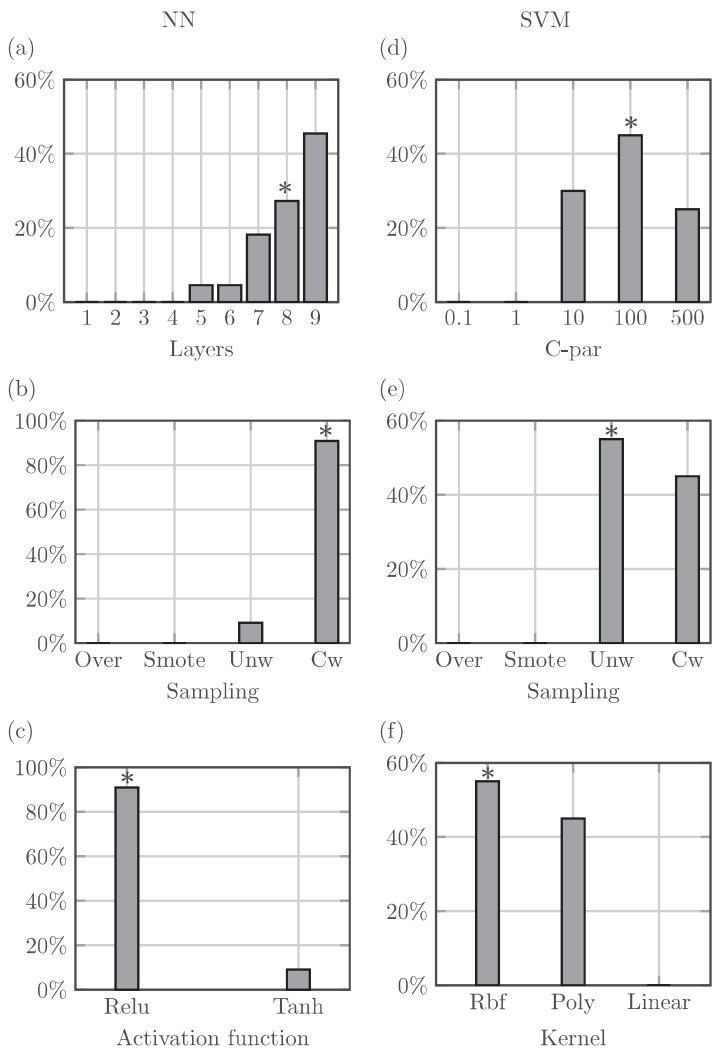

**Figure 13.** Properties of top 1% model configurations for the drought case. The stars denote the characteristics of the best-performing configurations according to the highest area under the precision-sensitivity curve.



**Table 1.** Main features of the selected (quasi-)global precipitation datasets.

| Dataset | Type | Resolution | Frequency | Coverage | Time span | Latency | Reference |
|---------|------|-----------|-----------|----------|-----------|---------|-----------|
| CCS | Satellite | 0.04° | 1h | 60°S - 60°N | January 2003 - present | 6h | Hong et al. (2004) |
| CHIRPS | Satellite-Gauge | 0.05° | 1d | 50°S - 50°N | January 1981 - present | 3 weeks | Funk et al. (2015) |
| CHIRP | Satellite | | | | | 3d | |
| CMORPH | Satellite-Gauge | 0.07° | 3h | 60°S - 60°N | January 1998 - present | 14 d | Joyce et al. (2004) |
| GSMaP | Satellite | 0.10° | 1h | 60°S - 60°N | March 2000 - present | 12h | Ushio and Kachi (2010) |
| IMERG | Satellite | 0.10° | 30min | 60°S - 60°N | June 2000 - present | 12h | Bolvin et al. (2018) |
| PERSIANN | Satellite | 0.25° | 1h | 60°S - 60°N | March 2000 - present | 48h | Sorooshian et al. (2000) |



**Table 2.** Main features of the selected soil moisture dataset.

| Dataset | Type | Resolution | Frequency | Coverage | Time span | Latency | Reference |
|---------|------|-----------|-----------|----------|-----------|---------|-----------|
| ERA5 | Reanalysis | 0.25° | 1h | Global | January 1979 - present | 5 days | ECMWF et al. (2018) |

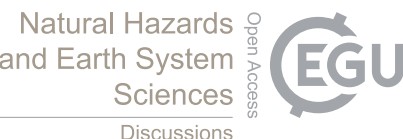

**Table 3.** Drought classification based on SPI according to McKee et al. (1993)

| Category | SPI | Probability (%) |
|---|---|---|
| Extremely wet | 2.00 and above | 2.3 |
| Severely wet | 1.50 to 1.99 | 4.4 |
| Moderately wet | 1.00 to 1.49 | 9.2 |
| Near normal | -0.99 to 0.99 | 68.2 |
| Moderately dry | -1.49 to -1.00 | 9.2 |
| Severely dry | -1.50 to -1.99 | 4.4 |
| Extremely dry | -2 and below | 2.3 |



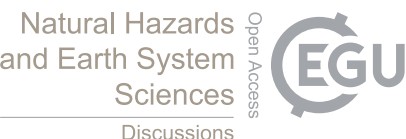

**Table 4.** Contingency table for the deterministic estimates of a series of binary events.

|  |  | Event Observed |  |
| --- | --- | --- | --- |
| Event predicted | Yes | No | Total |
| Yes | $a$ (True Positive or Hits) | $b$ (False Positive) | $a + b$ |
| No | $c$ (False Negative) | $d$ (True Negative) | $c + d$ |
| Total | $a + c$ | $b + d$ | $a + b + c + d = n$ |



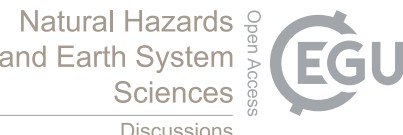
Table 5. Key metrics for the evaluation of model performance; a, b, c and d are defined in Table 4.

| Metric | Equation |
| --- | --- |
| Accuracy | $(a+d)/n$ |
| Precision | $a/(a+b)$ |
| Sensitivity (Recall) | $a/(a+c)$ |
| False alarm rate | $b/(b+d)$ |
| F1 score | $2*\frac{Precision*Sensitivity}{Precision+Sensitivity}$ |
| AUC under the ROC curve | $\int_0^1 ROC(t)dt$ |
| AUC under the PS curve | $\int_0^1 PS(t)dt$ |




**Table 6.** Breakdown of all the configuration explored by algorithm and type of hazard.

| Model | Parameter | Flood | Drought |
|---|---|---|---|
| NN | Input dataset combinations | 61 combinations of environmental variables | 61 combinations of environmental variables<br>4 SPI (1,3,6,12) |
| | Sampling | Unweighted<br>Class Weight<br>Over-sampling<br>SMOTE | Unweighted<br>Class Weight |
| | Loss | Binary Cross Entropy | Binary Cross Entropy |
| | Optimizer | ADAM | ADAM |
| | Number of layers & nodes | Layers: $[1;9]$<br>Nodes: $2^{nl+1}:2^{nl+9}(*)$ | Layers: $[1;9]$<br>Nodes $2^{nl+1}:2^{nl+9}(*)$ |
| | Activations | ReLu<br>Tanh | ReLu<br>Tanh |
| Number of Configurations | | 4392 | 8784 |
| SVM | Input dataset combinations | 67 combinations of environmental variables | 67 combinations of environmental variables<br>4 SPI (1,3,6,12) |
| | Sampling technique | Unweighted<br>Class Weight<br>Over-sampling<br>SMOTE | Unweighted<br>Class Weight |
| | C-Regularization parameter | $C=(0.1,1,10,100,500)$ | $C=(0.1,1,10,100,500)$ |
| | Kernel Function | Linear<br>Polynomial<br>Radial Basis | Linear<br>Polynomial<br>Radial Basis |
| Number of Configurations | | 4020 | 8040 |

$(*)$: nl: number of layers



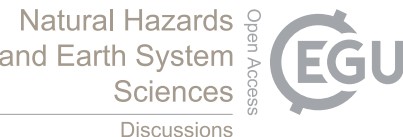

**Table 7.** Comparison metrics best configuration for flood by method.

| Method | Precision | Sensitivity | Specificity | F1 Score | Accuracy |
|--------|-----------|-------------|-------------|----------|----------|
| NN | 0.57 | 0.57 | 0.99 | 0.57 | 0.98 |
| SVM | 0.63 | 0.49 | 0.99 | 0.55 | 0.98 |
| LR | 0.46 | 0.42 | 0.99 | 0.43 | 0.97 |

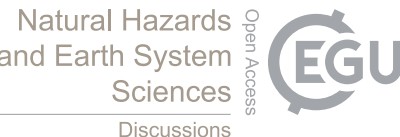

**Table 8.** Comparison metrics best configuration for drought by method.

| Method | Precision | Sensitivity | Specificity | F1 Score | Accuracy |
|--------|-----------|-------------|-------------|----------|----------|
| NN  | 0.95 | 1.00 | 0.99 | 0.97 | 0.99 |
| SVM | 0.96 | 0.96 | 0.99 | 0.96 | 0.98 |
| LR  | 0.63 | 0.74 | 0.89 | 0.68 | 0.85 |





Table A1. Past flood events in the Dominican Republic over the period from 2000 to 2019.

| Start date | End date | Duration d | Reference |
|---|---|---|---|
| 16/4/2003 | 18/4/2003 | 3 | ADRC and UNDRR (2020),CRED (2019) |
| 14/11/2003 | 14/11/2003 | 1 | ADRC and UNDRR (2020),CRED (2019) |
| 20/11/2003 | 24/11/2003 | 5 | Brakenridge (2002) |
| 6/12/2003 | 8/12/2003 | 3 | Brakenridge (2002) |
| 23/5/2004 | 25/5/2004 | 3 | ADRC and UNDRR (2020),CRED (2019) |
| 16/9/2004 | 18/9/2004 | 3 | ADRC and UNDRR (2020) |
| 23/10/2005 | 26/10/2005 | 4 | Brakenridge (2002) |
| 26/3/2007 | 30/3/2007 | 5 | Brakenridge (2002),CRED (2019) |
| 18/8/2007 | 21/8/2007 | 4 | OCHA (2020) |
| 28/10/2007 | 1/11/2007 | 5 | Brakenridge (2002),CRED (2019) |
| 11/12/2007 | 12/12/2007 | 2 | ADRC and UNDRR (2020),CRED (2019) |
| 15/8/2008 | 18/8/2008 | 4 | ADRC and UNDRR (2020),CRED (2019) |
| 23/1/2009 | 30/1/2009 | 8 | ADRC and UNDRR (2020),Brakenridge (2002) |
| 21/5/2009 | 25/5/2009 | 5 | ADRC and UNDRR (2020) |
| 15/2/2010 | 16/2/2010 | 2 | ADRC and UNDRR (2020),CRED (2019),Brakenridge (2002) |
| 22/6/2010 | 27/6/2010 | 6 | ADRC and UNDRR (2020),Brakenridge (2002) |
| 15/7/2010 | 24/7/2010 | 10 | CRED (2019) |
| 2/6/2011 | 7/6/2011 | 6 | ADRC and UNDRR (2020) |
| 4/8/2011 | 8/8/2011 | 5 | Brakenridge (2002),CRED (2019) |
| 23/4/2012 | 25/4/2012 | 3 | ADRC and UNDRR (2020),CRED (2019) |
| 25/8/2012 | 30/8/2012 | 6 | OCHA (2020) |
| 22/10/2012 | 30/10/2012 | 9 | OCHA (2020) |
| 23/8/2014 | 25/8/2014 | 3 | Brakenridge (2002) |
| 1/11/2014 | 6/11/2014 | 6 | Davies et al. (2008),CRED (2019) |
| 20/2/2015 | 21/2/2015 | 2 | ADRC and UNDRR (2020),CRED (2019),Davies et al. (2008) |
| 28/8/2015 | 29/8/2015 | 2 | OCHA (2020) |
| 7/5/2016 | 8/5/2016 | 2 | Brakenridge (2002),Davies et al. (2008) |
| 31/7/2016 | 2/8/2016 | 3 | Davies et al. (2008) |
| 2/10/2016 | 6/10/2016 | 5 | The International Charter Space and Major Disasters (2016) |
| 7/11/2016 | 15/11/2016 | 9 | Brakenridge (2002),CRED (2019) |
| 22/4/2017 | 25/4/2017 | 4 | Brakenridge (2002),CRED (2019),Davies et al. (2008) |
| 6/9/2017 | 7/9/2017 | 2 | OCHA (2020) |
| 20/9/2017 | 25/9/2017 | 6 | Davies et al. (2008) |
| 15/3/2018 | 20/3/2018 | 6 | Brakenridge (2002),Davies et al. (2008) |
| 4/5/2018 | 7/5/2018 | 4 | Davies et al. (2008) |



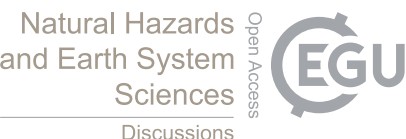

**Table A2.** Past drought events in the Dominican Republic over the period from 2000 to 2019.

| Start date | End date | Duration d | Reference |
|---|---|---|---|
| May 2000 | March 2003 | 1034 | Cornell University (2018) |
| October 2009 | April 2010 | 182 | Payano-Almanzar and Rodriguez (2018) |
| November 2013 | September 2014 | 304 | Payano-Almanzar and Rodriguez (2018) |
| April 2015 | January 2017 | 641 | Payano-Almanzar and Rodriguez (2018) |
| November 2018 | March 2019 | 120 | Global Disaster Alert and Coordination System (2018) |



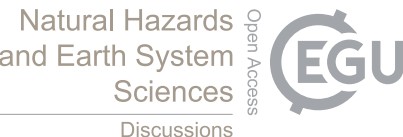

**Table A3.** NN metrics median value of the top 5% configuration according to F1 Score.

|       | Precision | Sensitivity | Specificity | F1 Score | Accuracy |
|-------|-----------|-------------|-------------|----------|----------|
| SPI1  | 0.7931    | 0.9000      | 0.9448      | 0.8387   | 0.9202   |
| SPI3  | 0.8163    | 0.8864      | 0.9581      | 0.8454   | 0.9336   |
| SPI6  | 0.9024    | 0.8919      | 0.9819      | 0.9167   | 0.9704   |
| SPI12 | 0.9423    | 0.9800      | 0.9868      | 0.9524   | 0.9751   |




**Table A4.** SVM metrics median value of the top 5% configuration according to F1 Score.

|       | Precision | Sensitivity | Specificity | F1 Score | Accuracy |
|-------|-----------|-------------|-------------|----------|----------|
| SPI1  | 0.8764    | 0.7915      | 0.9684      | 0.7709   | 0.9048   |
| SPI3  | 0.8977    | 0.8660      | 0.9744      | 0.8341   | 0.9300   |
| SPI6  | 0.9459    | 0.9629      | 0.9861      | 0.9317   | 0.9716   |
| SPI12 | 0.9532    | 0.9606      | 0.9856      | 0.9465   | 0.9751   |