# Peer review of "The potential of machine learning for weather index insurance"

_Natural Hazards and Earth System Sciences, 2020_

## Referee Comment (RC1) · Anonymous Referee #1 · 24 Aug 2020

The manuscript presents an assessment of two machine learning methods for weather index insurance. My overall impression is that this is a methodologically sound albeit not overly innovative study. It avoids many pitfalls that are sometimes overlooked even in peer-reviewed publications. The train - validation - test split is performed accurately, the problem of class imbalance is tackled adequately (using suitable performance metrics), and presentation quality in terms of figures is good. My recommendation is that this paper can be accepted subject to minor revisions.

I have some minor remarks on the contents and some technical remarks, which I have noted below.

**1 General Remarks**

- That the authors have put some effort into making the study is easily understandable for people not intimately familiar with machine learning methods. While this is commendable in a journal with a core focus on natural hazards, I feel that the manuscript is a bit lengthy at times. Some parts of sections 1 (Introduction) and particularly 2 (Methodology) could be shortened in order to make them more concise. Some parts read like an introductory book on machine learning. I suggest to go over section 2 again and streamline some of the rather basic parts.

- Please also double check the language throughout the text, especially syntax (e.g. line 289: *Hereinafter is proposed a procedure (...)*).

- I would refrain from using the term 'big data' in this context and adjust the title accordingly. Simply because the authors use larger data sets, this is not a novel big data problem per se.

**2 Specific remaks**

- Introduction

  – Line 15: I recommend to avoid the term 'significant' in a methodology-oriented paper. This might lead to confusion with respect to statistical significance.

  – Line 65 ff: This section states the core aim of the paper. Please add information on the input data source that is used. Currently, this essential statement is missing. In addition, I suggest to more precisely refer to flood and drought in this statement: '(...) is capable of objectively identifying and classifying

extreme *flood and drought* events *from satellite and gauge data products* in near-real time (...)'.

- Methodology

  – Line 129ff: These 5 criteria are important. I would welcome a reference of the actual values for these five criteria in the text. Maybe add a table featuring spatial and temporal metadata of the datasets used?

  – Line 325: Missing year in Mueller and Massaron

  – It is not ultimately clear which method for tackling class imbalance was used. I realized this when reading the results section, but the authors might want to add a sentence that this was also tuned as a model parameter in the methods section.

  – Since different methods for approaching class imbalance were used: Is there a reasons why procedures for undersampling were not considered? Or combinations such as SMOTE + undersampling?

  – Line 366f: Please check reference (M. and M.N., 2015)

  – Similar to the class imbalance method, it is unclear in the methods section which performance metrics have been used to compare the performance of the model. Was one specific metric used, or was the decision reached using all metrics presented in Tab. 5 by comparing all of them somehow? This is mentioned in the results section, but it is not clear when reading the methods section. I would argue that this is a methodological decision, not a result of the analysis.

  – The reference model on logistic regression is not ultimately clear. Did the authors use simple logisitic regression? Which link function was used? Did the authors include interaction effects? Did the authors use nonlinear effects? Simple logistic regression is fine as a reference model, but I think this could be stated more clearly.

- – Is there any particular justification why these two methods were selected specifically? My guess would be that a simple random forest with default parameters would probably perform equally well.

- Results and discussion

  - – I think more focus on the discussion would be beneficial. Results are described in this section, and findings are briefly commented. However, I am under the impression that there is some imbalance between the first half of the manuscript, which is quite extensive, and the discussion of the results, which is quite sparse. What have we learned from this study? Which novel aspects does this analysis show? What do the results mean for the Dominican Republic? Which impacts do the findings have on the study area?

---

## Referee Comment (RC2) · Anonymous Referee #2 · 23 Mar 2021

General comments

This paper explores whether satellite and reanalysis data for rainfall and soil moisture can be combined using machine learning methods to assess, in an objective way, whether floods or droughts are happening or have recently occurred. This is placed in the context of improving index insurance. The paper is extremely detailed in terms of how the machine learning models are constructed, and validation metrics.

1) My main comment is: the paper is very heavy on text-book style review of methods (which isn't a bad thing), and very heavy on technical detail (which isn't a bad thing), but lacks any exhibits that show clearly whether the methods actually work or not. There are masses of technical validation metrics. But what I personally would like to see are

some results along the lines of: a) we took the data shown in figure 7 (predicting this data is what the whole thing is about in the end) b) we split that data in half, trained the models on one half, chose the best model, and tested it on the other half c) and for the single best model, here's a picture that shows the results of that side by side with the actual floods and droughts that occurred in the validation period. Did it capture them all, or half of them, or none of them? d) then I'd be able to look at that and make a judgement as to whether the method works or not.

Specific comments

2) There's a whole discussion about training and validation data, but then in the end it's not clear how the data is actually split into training and validation data (relates to point 1 above), in relation to Figure 7. The construction of the validation is critical for us to be able to understand whether there's anything in this or not, especially since a large part of the scientific community associates the word 'machine learning' with 'overfitting', and will be sceptical.

3) With such a small amount of data, and after testing so many models and configurations (line 341: 'almost boundless domain of model configurations'), it seems to me that overfitting is quite likely. Could the authors elaborate on why testing so many configurations doesn't lead to overfitting? And if you are evaluating the models against each other using the validation dataset, of course one model will do best. How do we know that the model that does best would genuinely do best in a true out of sample sense? Don't you need another level of cross-validation?

4) Line 18 says \$3.3B. This is wrong by several orders of magnitude. Individual events during that period were in excess of \$50B (since at this point you are talking globally).

5) The word 'loss' is used with two different meanings, as far as I can tell. Line 105=loss in the usual sense of damages, vs line 249 in a technical sense. This is a bit confusing. Different terminology should be used, somehow, to avoid this.

6) I think it should be made clear that the runoff model – flood intensity relationships are simplistic relative to start of the art runoff and flood modelling as practised by hydrologists

7) Line 176 refers to loss data. What is this loss data?

8) Line 319, there is a comment that TensorFlow allows 'embedding the validation process into the construction of the model'. That sounds like overfitting to me. Please explain how this is consistent with the claim that the data is really being split in order to do out of sample validation.

9) Is reanalysis data really available soon enough to be useful? I thought it usually appears at least a year or two later, but maybe I'm wrong.

10) There should be a bit more discussion about the problems with satellite data and re-analyses (i.e., talk about the reasons why these data-sets aren't really used at present for index insurance purposes, even after 20 years of academics suggesting that they should be).

11) As far as I understand it, there has been no comparison here with standard methods for assessing whether an event has occurred, which are based on rain gauges, levels of river flow, etc. That should be pointed out.

12) Are there any further diagnostics that could be produced to help show that the model is really doing something sensible, to help allay the suspicion that some readers may have that it's all just over-fitted?

Technical corrections

line 171: you say $T\_t$, but don't you mean $Y\_t$?

line 205: SP1, 3 etc need to be defined. I can guess what they are, but they should be defined.

line 367: is that citation really correct? Is the person's name just M?

i.e. and e.g. are usually followed by commas I believe

the plural of reanalysis is reanalyses
* * *
Interactive
comment

---

## Author Comment (AC1) · 17 Apr 2021

**Introduction**

**C:***The manuscript presents an assessment of two machine learning methods for weather index insurance. My overall impression is that this is a methodologically sound albeit not overly innovative study. It avoids many pitfalls that are sometimes overlooked even in peer-reviewed publications. The train - validation - test split is performed accurately, the problem of class imbalance is tackled adequately (using suitable performance metrics), and presentation quality in terms of figures is good. My recommendation is that this paper can be accepted subject to minor revisions.*

**R:** Dear Reviewer,

Thank you very much for your time and effort reviewing our manuscript. This response (R) carefully addresses all the comments (C). Where deemed appropriate, modifications to the manuscript are proposed (red underlined text indicates additions to the manuscript, blue strikethrough text indicates removed text).

**General Remarks**

**C:** *That the authors have put some effort into making the study is easily understandable for people not intimately familiar with machine learning methods. While this is commendable in a journal with a core focus on natural hazards, I feel that the manuscript is a bit lengthy at times. Some parts of sections 1 (Introduction) and particularly 2 (Methodology) could be shortened in order to make them more concise. Some parts read like an introductory book on machine learning. I suggest to go over section 2 again and streamline some of the rather basic parts.*

**R:** We agree that parts of Section 2 may be streamlined. We propose the following changes, which we believe will improve the readability of the manuscript while still providing some of the key concepts of the methods that are used.

At line 215:

[revised manuscript text omitted]

 how class imbalance might affect measures that use true negative in their computation."

**C:***Please also double check the language throughout the text, especially syntax (e.g. line 289: Hereinafter is proposed a procedure (...)).*

**R:** In the revised manuscript we will carefully check and improve the writing and syntax, as suggested.

**C:** *I would refrain from using the term 'big data' in this context and adjust the title accordingly. Simply because the authors use larger data sets, this is not a novel big data problem per se.*

**R:** We agree and propose to change the title to:

"The potential of machine learning for weather index insurance"

**Specific Remarks**

**C:** *Line 15: I recommend to avoid the term 'significant' in a methodology- oriented paper. This might lead to confusion with respect to statistical significance.*

**R:** We agree and will replace the word "significant" with "substantial" here.

**C:** *Line 65 ff: This section states the core aim of the paper. Please add information on the input data source that is used. Currently, this essential statement is missing. In addition, I suggest to more precisely refer to flood and drought in this statement: '(...) is capable of objectively identifying and classifying extreme flood and drought events from satellite and gauge data products in near-real time (...)'.*

**R:** We suggest the following change:

"we propose and apply a machine learning methodology that is capable of objectively identifying and classifying extreme weather events, namely flood and drought, in near-real time, using quasi-global gridded climate datasets derived from satellite imagery or a combination of observations and satellite imagery. This methodology is then used to address the following research questions"

**C:** *Line 129ff: These 5 criteria are important. I would welcome a reference of the actual values for these five criteria in the text. Maybe add a table featuring spatial and temporal metadata of the datasets used?*

**R:** We agree. We would like to point out that Table 1 and Table 2 already report the information regarding the 5 criteria listed. An effort to highlight the tables will be made in the article adding a reference. We suggest the following change to the manuscript to make the connection more direct.

"... Based on a comprehensive review of available datasets, we found six rainfall datasets and one soil moisture dataset, comprising 4 layers, matching the above criteria. The main features of the selected datasets are reported in Table 1 and Table 2..."

**C:** *Line 325: Missing year in Mueller and Massaron*

**R:** The reference will be corrected to: "(Mueller and Massaron, 2016)"

**C:** *It is not ultimately clear which method for tackling class imbalance was used. I realized this when reading the results section, but the authors might want to add a sentence that this was also tuned as a model parameter in the methods section.*

**R:** We agree with the reviewer that this should be made more clear. In the manuscript, we tried to specify this aspect at line 342 using the term "data augmentation technique", which we reckon might create some confusion. We propose the following change at line 342:

"... all the  resampling techniques previously introduced were tested…"

**C:** *Since different methods for approaching class imbalance were used: Is there a reasons why procedures for undersampling were not considered? Or combinations such as SMOTE + undersampling?*

**R:** When evaluating which techniques were more suitable to tackle class imbalance, we concluded that using undersampling would have reduced our datasets to such a dimension that was not deemed appropriate for the training of the ML algorithm. Accordingly, for the same reason, a combination of SMOTE and undersampling was not used. Since SMOTE generates synthetic samples "close" to the sample it is trying to replicate, undersampling from a group built as such could lead to the loss of some real events.

We propose to add the following sentence at line 335 to clarify:

"(…) Oversampling, SMOTE and class weight were the resampling techniques deemed more appropriate to the scope of this work, namely, identifying events in the minority class. (…)"

**C:** *Line 366f: Please check reference (M. and M.N., 2015)*

**R:** Will be corrected in "(Hossin and Sulainman, 2015)"

**C:** *Similar to the class imbalance method, it is unclear in the methods section which performance metrics have been used to compare the performance of the model. Was one specific metric used, or was the decision reached using all metrics presented in Tab. 5 by comparing all of them somehow? This is mentioned in the results section, but it is not clear when reading the methods section. I would argue that this is a methodological decision, not a result of the analysis.*

**R:** Thank you for pointing out this missing information. We agree that this is a methodological decision and we'd like to propose the following changes at line 402:

"Lastly, once the domain of all configurations is well established and the best settings of the ML algorithms were selected based on the highest values of F1 score and area under the PS curve , the predictive performances of the models are compared to those of logistic regression (LR) models. (…)"

**C:** *The reference model on logistic regression is not ultimately clear. Did the authors use simple logisitic regression? Which link function was used? Did the authors include interaction effects? Did the authors use nonlinear effects? Simple logistic regression is fine as a reference model, but I think this could be stated more clearly.*

**R:** We suggest the following addition to the manuscript to clarify the doubts raised regarding the properties of the logistic regression

" The logistic regression adopted as a baseline takes as input multiple environmental variables, in line with the procedure followed for the ML methods and used a logit function (eq.6) as link function, neglecting interaction and nonlinear effects amid predictors. The logistic regression is a more traditional statistical model whose application to index insurance has recently been proposed, and can be said to already represent in itself an improvement over common practice in the field (Calvet et al., 2017; Figueiredo et al., 2018)."

**C:** *Is there any particular justification why these two methods were selected specifically? My guess would be that a simple random forest with default parameters would probably perform equally well.*

**R:** We limited our analyses to these two types of ML methods for the sake of brevity. We reckon that other ML methods might enable comparable results. We propose adding a comment about this in the Conclusion:

"It is also worth noting that although this work focuses on the application of neural network and support vector machine models, we expect that comparable results could be obtained using other machine learning algorithms, which calls for further research."

**C:** *I think more focus on the discussion would be beneficial. Results are described in this section, and findings are briefly commented. However, I am under the impression that there is some imbalance between the first half of the manuscript, which is quite extensive, and the discussion of the results, which is quite sparse. What have we learned from this study? Which novel aspects does this analysis show? What do the results mean for the Dominican Republic? Which impacts do the findings have on the study area?*

**R:** In agreement with what was discussed in the general remarks section about the length of the methodology section, an effort was made to slim down the introductory part and to improve the results and discussion section, to strike a better balance between the two parts. In an effort to provide also an answer to the reviewer's questions, we propose the following additions in the conclusion section along with the cuts already mentioned for Section 2:

At line 599:

"It is also worth noting that although this work focuses on the application of neural network and support vector machine models, we expect that comparable results could be obtained using other machine learning algorithms, which calls for further research."

We found appropriate moving the following paragraph from the methodology section to conclusion:

"Also, while here we focus on performance-based evaluation measures, an alternative approach may be to quantify the utility of the predictive systems. By taking into account actual user expenses and thus specific weights for different model outcomes, a utility-based approach may potentially lead to different decisions regarding model selection and definition of the trigger threshold (Murphy and Ehrendorfer, 1987; Figueiredo et al., 2018). This aspect is outside the scope of the present article and warrants further research."

At line 601:

"Although several issues raised in this article warrant further research, there is clear potential in the application of machine algorithms to take advantage of increasing amounts of available environmental data within the context of weather index insurance. The capability of these algorithms to reduce basis risk with respect to traditional methods could play a key role in the adoption of parametric insurance in the Dominican context and more generally for those countries that detain a low level of information about risk. Indeed, being able to rely on global data that are disentangled from the resources of a given territory, both from the point of view of climate data (e.g., lack of rain-gauge network) and from the point of view of information about past natural disasters, is an appealing feature of the work presented that would make the

approach proposed feasible for other countries. The framework presented and topics discussed in this study provide a scientific basis for the development of robust and operationalizable parametric risk transfer products."

---

## Author Comment (AC2) · 17 Apr 2021

**General comments**

**C:** *This paper explores whether satellite and reanalysis data for rainfall and soil moisture can be combined using machine learning methods to assess, in an objective way, whether floods or droughts are happening or have recently occurred. This is placed in the context of improving index insurance. The paper is extremely detailed in terms of how the machine learning models are constructed, and validation metrics.*

**R:** Dear Reviewer,

Thank you very much for your time and effort reviewing our manuscript. This response (R) carefully addresses all the comments (C). Where deemed appropriate, modifications to the manuscript are proposed (red underlined text indicates additions to the manuscript, blue strikethrough text indicates removed text).

**C:** *My main comment is: the paper is very heavy on text-book style review of methods (which isn't a bad thing), and very heavy on technical detail (which isn't a bad thing), but lacks any exhibits that show clearly whether the methods actually work or not. There are masses of technical validation metrics. But what I personally would like to see are some results along the lines of: a) we took the data shown in figure 7 (predicting this data is what the whole thing is about in the end) b) we split that data in half, trained the models on one half, chose the best model, and tested it on the other half c) and for the single best model, here's a picture that shows the results of that side by side with the actual floods and droughts that occurred in the validation period. Did it capture them all, or half of them, or none of them? d) then I'd be able to look at that and make a judgement as to whether the method works or not.*

**R:**
After analysing this comment together with the rest of the review, our understanding is that the Reviewer has some concerns regarding the performances of the models, which originate in part from how he/she perceives that the validation process was carried out. Upon critically reviewing the original manuscript, we believe that this is likely because the validation process was not sufficiently well explained. More specifically, Section 2.3.3 describes in detail which are the best practices used in the validation of a machine learning model without stating clearly what was done in our work. Hence, we suggest the following change at line 318 when we describe what was done in our work regarding the splitting and validation, which we hope clarifies this aspect.

"In this work, the proper training of the NN was exerted splitting the dataset in 3 parts: training (60%), validation (15%) and testing (25%) set. During training, the neural network used only the training set, evaluating the loss on the validation set at each iteration of the training process. After the training, the performance of the model was evaluated on the testing set that the model has never seen. Concerning the SVM, a k-fold cross validation (Mosteller and Tukey, 1968) was used to validate the  model, using 5 folds created by preserving the percentage of

sample of each class, the algorithm was therefore trained on 80% of the data and its performances were evaluated on 20% of the remaining data that the model has never seen whilst for NN, TensorFlow allows the user to declare a percentage of the data that is retained as validation data at each iteration of the training loop, therefore, embedding the validation process into the construction of the model."

We hope that the above changes clarify how the splitting of the dataset and the training process were performed, which is a central aspect for the construction of the models and the robust evaluation of their performances.

Going back to the Reviewer's comment more specifically, we would like to highlight that the objective of verification metrics such as the ones we used in our manuscript is summarizing the overall prediction quality of a set of predictions - which is the same as to "assess whether the methods actually work or not". Notwithstanding, we do recognize that looking at an actual plot of observations versus models predictions can sometimes be an easier way to have a feeling for how the model is performing and therefore we considered adding such a plot to the article. Below is a figure depicting, for the days and weeks belonging only to the testing set (i.e., data that the models have never seen before), observed flood and drought events and the corresponding predictions from the three methods discussed in the manuscript.

[Figure]

*Figure 1: Comparison of prediction over the testing set of the three methods. Flood Case.*

[Figure]

*Figure 2: Comparison of prediction over the testing set of the three methods. Drought Case.*

The above plot is essentially a visual representation of what the metrics reported in Table 7 and 8 report. The question is whether adding such plots would objectively improve the article. After careful consideration, we believe that while plots such as these may be more visually appealing, they do not significantly contribute toward a better interpretation of the results. Therefore, we propose to keep the numerical representation, which we believe provides an easier way to compare results among methods and is able to summarize in a short and quantitative way the results, while including the above plots as Supplementary material.

**Specific comments**

**C:** *There's a whole discussion about training and validation data, but then in the end it's not clear how the data is actually split into training and validation data (relates to point 1 above), in relation to Figure 7. The construction of the validation is critical for us to be able to understand whether there's anything in this or not, especially since a large part of the scientific community associates the word 'machine learning' with 'overfitting', and will be sceptical.*

**R:** We agree that proper validation is crucial and paid careful attention to its construction. We hope the changes suggested above provide a good improvement to the reading experience and dispel any doubt about the partitioning of the dataset.

**C:** *With such a small amount of data, and after testing so many models and configurations (line 341: 'almost boundless domain of model configurations'), it seems to me that overfitting is quite likely. a)Could the authors elaborate on why testing so many configurations doesn't lead to overfitting? b) And if you are evaluating the models against each other using the validation dataset, of course one model will do best. How do we know that the model that does best would genuinely do best in a true out of sample sense? Don't you need another level of cross-validation?*

**R:** I'll address the reviewer questions in order:
   a) The different model configurations tested do not share the training process. Each model, initialized with different parameters and configurations the way we presented in the manuscript, has its own training process, hence, the training of one configuration does not influence the others. We would like to emphasize that having different model configurations to train is not to overfitting, which by definition refers to the ability of a model to reproduce predictions or analysis valid only for a particular set of data. This is prevented in our work by providing an accurate splitting of the data, as described above, and by taking other measures during the training of the models that are recalled later on in this response.
   **b)** The model performances reported were obtained from the testing partition of the data that we consider a "true out of sample". The definition of the testing dataset will be

improved and clarified with the text addition at line 318 mentioned above. This concept is recalled in the result section at line 473.

**C:** *Line 18 says $3.3B. This is wrong by several orders of magnitude. Individual events during that period were in excess of $50B (since at this point you are talking globally).*

**R:** We will change the number to the correct amount, which is $3300B. (Hoeppe, 2016)

**C:** *The word 'loss' is used with two different meanings, as far as I can tell. Line 105=loss in the usual sense of damages, vs line 249 in a technical sense. This is a bit confusing. Different terminology should be used, somehow, to avoid this.*

**R:** We understand that the usage of loss might create some confusion, but we also acknowledge that the term loss in these two contexts has a specific meaning that would be lost changing the terminology. Therefore, we suggest introducing the acronym LF defining the loss function throughout the rest of the manuscript.

**C:** *I think it should be made clear that the runoff model – flood intensity relationships are simplistic relative to start of the art runoff and flood modelling as practised by hydrologists*

**R:** Complex state-of-the-art models are typically not considered in parametric insurance products, as such products are meant to be based on simple indices obtained directly from environmental variables without the need for large modelling efforts that typically require much larger amounts of data. Even if this is somewhat implied, we agree that the simplified approach should be pointed out and therefore propose the following addition to highlight this aspect without going into the details of the runoff/flood modelling practices, which we believe are beyond the scope of the manuscript:
At line 157:
"To achieve this, we adopt a variable transformation to emulate, in a simplified manner, the physical processes behind the occurrence of flood damage due to rainfall, (…)."

**C:** *Line 176 refers to loss data. What is this loss data?*

**R:** In this instance we refer to reported occurrence of loss data. We understand the writing might be a bit troublesome, thus, we suggest the following change at line 176 to avoid any misunderstanding, recalling terminology introduced before in the manuscript and used throughout the whole paper:

"...fitting a logistic regression model to concurrent potential flood intensity and reported occurrences of losses caused by flood events data, and maximizing the likelihood using a quasi-Newton method:"

**C:** *Line 319, there is a comment that TensorFlow allows 'embedding the validation process into the construction of the model'. That sounds like overfitting to me. Please explain how this is consistent with the claim that the data is really being split in order to do out of sample validation.*

**R:** We agree that this statement may be somewhat unclear and have proposed to replace it, as described above. Furthermore, we have made several proposals for improvement regarding model validation, which we hope clarify this topic.
Nevertheless, we would still like to provide information about the statement originally reported in the manuscript. When training the model, the function that is called in the code to carry out the training allows, among its parameters, to indicate the way you want to perform the validation. You can either directly pass the validation dataset to the function or declare the percentage of data you want to retain from the training set as part of the validation. In both cases, the validation datasets are not used during the training of the model but to evaluate the loss function during the training.

**C:** *Is reanalysis data really available soon enough to be useful? I thought it usually appears at least a year or two later, but maybe I'm wrong.*

**R:** The reanalysis data we used in our work are updated daily with a latency of about 5 days, as reported in table 2 and by the Copernicus documentation (Copernicus ERA-5)

**C:** *There should be a bit more discussion about the problems with satellite data and re-analyses (i.e., talk about the reasons why these datasets aren't really used at present for index insurance purposes, even after 20 years of academics suggesting that they should be).*

**R:** While some limitations are discussed in the introduction, we recognize that highlighting more about the shortcomings of using this type of data could be beneficial to the manuscript.

Addition to the manuscript at line 48:

"Satellite images are often available with high spatial resolution, but records are still short, with a maximum duration of around 30 years. Reanalysis, on the other hand, provides longer time series but tends to have a coarser spatial resolution. Moreover, satellite data should be checked for consistency with ground measurement which is not always feasible when the network of ground instruments is inadequate or non-existent (Loew et al., 2017). Although using satellite data has its own limitations, various index-based insurance products, exploiting remote-sensing data and reanalysis, have been developed in data sparse regions such as Africa and Latin America (Awondo, 2018; African Union, 2021;The World Bank, 2008). The combined use of various sources of information to detect the occurrence of extreme events is

valuable, since it can significantly improve the ability to correctly detect extreme events (Chiang et al., 2007) and a proper index design helps addressing the limitations brought by satellite data, as underlined in Black (2016).

**C:** *As far as I understand it, there has been no comparison here with standard methods for assessing whether an event has occurred, which are based on rain gauges, levels of river flow, etc. That should be pointed out.*

**R:** The historical event catalogue contains information about whether events have occurred or not. Standard methods, which we presume refer to threshold-based methods based on rain or stream gauges, are not applicable in this case as the availability of such data in the Dominican context is scarce. Nevertheless, note that in parametric insurance the use of such data is not optimal, as described in the Introduction.

**C:** *Are there any further diagnostics that could be produced to help show that the model is really doing something sensible, to help allay the suspicion that some readers may have that it's all just over-fitted*

**R:** We hope the discussion and clarification throughout this response have alleviated some of the concerns the review raised regarding overfitting. Nonetheless, we would like to summarize the measures taken in our work to prevent overfitting that are treated in the paper:
-   At line 253, we discuss the role of monitoring the training and validation in avoid overfitting and how countermeasures to stop the training can be taken whereas is needed. In practice, the training is stopped with a TensorFlow's call back called "EarlyStopping" that stops the training when a monitored metric, in our case the loss function, has stopped improving. We mentioned in the manuscript how this lack of improvements might very well be unhealthy for the model therefore granting the stop of the training.
-   The second paramount action taken against overfitting is the splitting of the data. We clarified how the splitting was done in the previous answers, here we would like to reiterate that the results presented were obtained from data the the model had never seen.

**Technical corrections**

**C:** *you say T_t, but don't you mean Y_t?*
**R:** Checked and corrected

**C:** *SP1, 3 etc need to be defined. I can guess what they are, but they should be defined.*

**R:** We agree with the reviewer that the terminology used should be explained clearly. We suggest the following addition to the manuscript:

"In this study SPI1, SPI3, SPI6 and SPI12 were computed, where the numeric values in the acronym refer to the period of accumulation in months (e.g. SPI3 indicates the standard precipitation index computed over a three months accumulation period)."

**C:** *is that citation really correct? Is the person's name just M?*
**R:** Will be corrected in "(Hossin and Sulainman, 2015)"

**C:** *i.e. and e.g. are usually followed by commas I believe*
**R:** Will be corrected where needed

**C:** *the plural of reanalysis is reanalyses*
**R:** Will be corrected where needed

**References**

African Union. (2021). *African Risk Capacity: Transforming disaster risk management & financing in Africa*. https://www.africanriskcapacity.org/

Awondo, S. N. (2018). SC. *Journal of Development Economics*. https://doi.org/10.1016/j.jdeveco.2018.10.004

Loew, A., Bell, W., Brocca, L., Bulgin, C. E., Burdanowitz, J., Calbet, X., Donner, R. V., Ghent, D., Gruber, A., Kaminski, T., Kinzel, J., Klepp, C., Lambert, J. C., Schaepman-Strub, G., Schröder, M., & Verhoelst, T. (2017). Validation practices for satellite-based Earth observation data across communities. *Reviews of Geophysics*, *55*(3), 779–817. https://doi.org/10.1002/2017RG000562

The World Bank. (2008). *Operational Innovations: Providing Immediate Funding After Natural Disasters*.

---

## Author Response (AR2)

**Reviewer 1**

*"Upon reading the revised manuscript again, I wondered if the expectation raised by the title (specifically the term "potential") as well as the introduction (c.f. the three research question posed in the intro) is sufficiently reflected in the conclusion. The outcome that SVM and NN perform better than logistic regression is not surprising. The authors could consider summarizing the actual potential of ML methods in the context of weather index insurance more concisely in the conclusion."*

**R:** Dear reviewer,

We would like to thank you for the time and effort spent during this second round of review. We took into consideration your final comment and tried to address more directly the concern raised elaborating more in detail in the conclusion at line 616:

"Although several issues raised in this article warrant further research, there is clear potential in the application of machine  learning algorithms in the context of weather index insurance. The first reason for this is strictly linked to the performances of the models. Indeed, the capability of these algorithms to reduce basis risk with respect to traditional methods could play a key role in the adoption of parametric 620 insurance in the Dominican context and more generally for those countries that  possess a low level of information about risk.  The second aspect, perhaps the most intriguing from the weather index insurance point of view, regards the ability of these algorithms to utilise and improve their performances using a growing amount of information (i.e., increasing the number of input variables). Indeed, the significant advances in data collection and availability observed in the last decades (i.e., improved instruments, more satellite missions, open access to data store services) made it so that vast amount of data are readily and freely available on a daily basis. Being able to rely on global data that are disentangled from the resources of a given territory, both from the point of view of climate data (e.g., lack of rain-gauge networks) and from the point of view of information about past natural disasters, is an  important feature of the work presented that would make the  proposed approach feasible and appealing for other countries.  Furthermore, similar technological improvements might be expected in the further development of machine learning algorithms. The scientific evolution of these models, and the possibility of establishing a pipeline that automatically and objectively trains the algorithm over time with updated and improved data (always allowing the monitoring of the process), are other appealing features of these kind of models. In conclusion, the framework presented and topics discussed in this study provide a scientific basis for the development of robust and  operationalisable ML-based parametric risk transfer products."

**Reviewer 2**

*"Thanks for responding to all my questions, and for making the two new figures (C1 and C2). I just have one comment: my view is that the paper would be greatly improved by including those figures in the main text. They are much easier to understand than Tables 7 and 8, especially for people who don't think about sensitivity and specificity every day. But I'll leave the final decision on that up to you"*

**R:** Dear reviewer,

We would like to thank you for the time and effort spent during this second round of review. Upon your suggestion, we decided to include figure C1 and C2 into the paper in the result section in order to provide a better reading experience.